# The Minimal Search Space for Conditional Causal Bandits

## Abstract

Causal knowledge can be used to support decision-making problems. This has been recognized in the causal bandits literature, where a causal (multi-armed) bandit is characterized by a causal graphical model and a target variable. The arms are then interventions on the causal model, and rewards are samples of the target variable. Causal bandits were originally studied with a focus on hard interventions. We focus instead on cases where the arms are *conditional interventions*, which more accurately model many real-world decision-making problems by allowing the value of the intervened variable to be chosen based on the observed values of other variables. This paper presents a graphical characterization of the minimal set of nodes guaranteed to contain the optimal conditional intervention, which maximizes the expected reward. We then propose an efficient algorithm with a time complexity of $O(|V|+|E|)$ to identify this minimal set of nodes. We prove that the graphical characterization and the proposed algorithm are correct. Finally, we empirically demonstrate that our algorithm significantly prunes the search space and substantially accelerates convergence rates when integrated into standard multi-armed bandit algorithms.

## 1. Introduction

Lattimore et al. (2016) introduce a class of problems termed *causal bandit* problems, where actions are interventions on a causal model, and rewards are samples of a chosen reward variable $Y$ belonging to the causal model. They focus on hard interventions, where the intervened variables are set to specific values, without considering the values of any other variables. We will refer to this as a hard-intervention causal bandit problem. They propose a best-arm identification al-

gorithm that utilizes observations of the non-intervened variables in the causal model to accelerate learning of the best arm as compared to standard multi-armed bandit (MAB) algorithms. Causal bandits have applications across a broad range of domains, particularly in scenarios requiring the selection of an intervention on a causal system. These include computational advertising and context recommendation (Bottou et al., 2013; Zhao et al., 2022), biochemical and gene interaction networks (Meinshausen et al., 2016; Basharin, 1959), epidemiology (Joffe et al., 2012), and drug discovery (Michoel and Zhang, 2023).

Most of the work in causal bandits (see Section 7) focuses on developing MAB algorithms which incorporate knowledge about the causal graph.

Lee and Bareinboim (2018), in contrast, use the fact that the causal graph is known not to develop yet another MAB algorithm, but to reduce the set of nodes (*i.e.* variables) of the causal graph on which hard interventions should be examined, thereby reducing the search space for hard-intervention causal bandit problems. In more detail, they define the SCM-MAB problem, where the agent has access to the causal graph $G = (\mathbf{V}, E)$ of a structural causal model (SCM) and wants to maximize a target variable $Y \in \mathbf{V}$ by playing arms which are *hard* interventions on *subsets of* $\mathbf{V}$. Their search space reduction algorithm identifies the set of all (minimal) subsets $\mathbf{X}$ of $\mathbf{V}$ such that there exists an SCM with graph $G$ for which some hard, multi-node intervention $do(\mathbf{X} = \mathbf{x})$ has maximal $\mathbb{E}_{Y \sim p_Y^{do(\mathbf{X} = \mathbf{x})}}[Y]$. Lee and Bareinboim (2019) and Lee and Bareinboim (2020) extend the approach of Lee and Bareinboim (2018) to the cases involving non-manipulable variables and mixed policies, respectively (see Section 7).

It is recognized in the MAB literature that, for many if not most applications, actions are taken in a context, that is, with available information (Lattimore and Szepesvári, 2020; Agarwal et al., 2014; Dudik et al., 2011; Jagerman et al., 2020; Langford and Zhang, 2007). *E.g.*, content recommendation based on the user's demographic characteristics, such as age, gender, nationality and occupation. Similarly, in causality, conditional interventions — where a variable $X$ is set to a value $g(\mathbf{Z})$ through some policy $g$ after observing other variables (a context) $\mathbf{Z}$ — are more realistic than hard

[1]Anonymous Institution, Anonymous City, Anonymous Region, Anonymous Country. Correspondence to: Anonymous Author <anon.email@domain.com>.

Preliminary work. Under review by the International Conference on Machine Learning (ICML). Do not distribute.

or soft[1]interventions in many real-world scenarios. Conditional interventions were first introduced in Pearl (1994) based on the argument that "In general, interventions may involve complex policies in which a variable $X$ is made to respond in a specified way to some set $\mathbf{Z}$ of other variables." Shpitser and Pearl (2012) motivate their interest in conditional interventions by providing the concrete example of a doctor selecting treatments based on observed symptoms and medical test results $\mathbf{Z}$ to improve the patient's health condition. The doctor performs interventions of the form $do(X_i = x_i)$, but "the specific values of the treatment variables are not known in advance, but instead depend on symptoms and test results performed 'on the fly' via policy functions $g_i$". Although the practical motivation for conditional interventions is clear, causal bandits with conditional interventions have not yet been studied, possibly due to the complexity of these interventions compared to the mathematically simpler hard and soft interventions.

**Novelty and Contributions:** This work, like that of Lee and Bareinboim (2018), uses the causal graph to reduce the search space of an MAB problem. Our work is novel because we consider the case where (i) the arms are *conditional interventions* (which generalize both hard and soft interventions); and (ii) the interventions are *single-node interventions*. This is the first time the minimal search space for a causal bandits problem with non-hard interventions is fully characterized. Such a characterization has also not been done for single-node interventions (of any kind). Our contributions are as follows: (a) we establish a graphical characterization of the minimal set of nodes guaranteed to contain the optimal node on which to perform a conditional intervention; and (b) we propose an algorithm which finds this set, given only the causal graph, with a time complexity of $O(|V| + |E|)$. As a supplementary result, we also show that, perhaps surprisingly, the exact same minimal set would hold for the optimization problem of selecting an atomic (*i.e.* single-node and hard) intervention in a deterministic causal model. We provide proofs for the graphical characterization and correctness of the algorithm, as well as experiments that assess the fraction of the search space that can be expected to be pruned using our method, in both randomly generated and real-world graphs, and demonstrate, using well-known real-world models, that our intervention selection can significantly improve a classical MAB algorithm.

All proofs of the results presented in the paper can be found in the appendix. The code repository containing the experiments can be found in the file submitted alongside the paper.

---

[1]In a soft intervention, the intervened variable keeps its direct causes (Peters et al., 2017).

## 2. Preliminaries

**Graphs and Causal Models**   We will make use of Directed Acyclic Graphs (DAGs). The main concepts of DAGs and notation used in this paper are reviewed in Appendix A. Furthermore, we operate within the Pearlian graphical framework of causality, where causal systems are modeled using Structural Causal Models (SCMs) (Peters et al., 2017; Pearl, 2009). An *SCM* $\mathfrak{C}$ is a tuple $(\mathbf{V}, \mathbf{N}, \mathcal{F}, p_{\mathbf{N}})$, where $\mathbf{V} = (V_1, \ldots, V_n)$ and $\mathbf{N} = (N_{V_1}, \ldots, N_{V_n})$ are vectors of random variables. The exogenous variables are pairwise independent, and are distributed according to the *noise distribution* $p_{\mathbf{N}}$, while each endogenous variable $V_i$ is a deterministic function $f_{V_i}$ of its noise variable $N_{V_i}$ and a (possibly empty) set of other endogenous variables $\mathrm{Pa}(V_i) \setminus \{V_i\}$, called the (proper) parents of $V_i$. The $V_i$ and $N_{V_i}$ are called *endogenous* and *exogenous* (or *noise*) variables, respectively. $R_V$ denotes the range of the random variable $V$. $\mathcal{F}$ is a set of functions $f_{V_i} : R_{\mathrm{Pa}(V_i)} \times R_{N_{V_i}} \to R_{V_i}$, termed *structural assignments*. The endogenous variables together with $\mathcal{F}$ characterize a DAG called the *causal graph* $G^{\mathfrak{C}} := (\mathbf{V}, E)$ of $\mathfrak{C}$, whose edge set is $E = \{(P, X) : X \in \mathbf{V}, P \in \mathrm{Pa}(X) \setminus \{X\}\}$. We denote by $\mathfrak{C}(G)$ the set of SCMs whose causal graph is $G$. Having an SCM allows us to model interventions: intervening on a variable changes its structural assignment $f_X$ to a new one, say $\tilde{f}_X$. This intervention is then denoted $do(f_X = \tilde{f}_X)$. In the simplest type of interventions, called *atomic interventions*, a variable $X$ is set to a chosen value $x$, thus replacing the structural assignment $f_X$ of $X$ with a constant function setting it to $x$. Such an intervention is denoted $do(X = x)$, and the SCM resulting from performing this intervention is denoted $\mathfrak{C}^{do(X=x)}$. The joint distribution over the endogenous variables resulting from the atomic intervention $do(X = x)$ is denoted $p^{do(X=x)}$ and called the *post-intervention distribution* for this intervention. Each realization $\mathbf{n} \in R_{\mathbf{N}}$ of the noise variables will be called a *unit*. A *deterministic SCM* is an SCM for which the noise distribution is a point mass distribution with all its mass on some (known) unit $\mathbf{n} \in R_{\mathbf{N}}$. Finally, nodes are denoted by upper case letters, sets of nodes by boldface letters, and variable values by lower case letters.

**Unrolled Assignments**   We will make use of the fact that the structural assignments of the ancestors of an endogenous variable $X$ (including its own structural assignment) can be composed to express $X$ as a function $\bar{f}_X(\mathbf{n})$ of the vector $\mathbf{n}$ of exogenous variables values. We call this the *unrolled assignment* of $X$. The formal definition can be found in Appendix B.

**Conditional Interventions**   Given an SCM $\mathfrak{C} = (\mathbf{V}, \mathbf{N}, \mathcal{F}, p_{\mathbf{N}})$ with causal graph $G$, $X \in \mathbf{V}$ and $\mathbf{Z}_X \subseteq \mathbf{V} \setminus \{X\}$, the *conditional intervention on $X$ given $\mathbf{Z}_X$ for the policy* $g : R_{\mathbf{Z}_X} \to R_X$, denoted $do(X = g(\mathbf{Z}_X))$, is the

intervention where the value of $X$ is determined by that of $\mathbf{Z}_X$ through $g$ (Pearl, 2009). The exact conditioning set $\mathbf{Z}_X$ for each $X$ will depend on the specific application. In order to study conditional interventions, we will need to make some assumptions of what nodes can reasonably be in $\mathbf{Z}_X$, *i.e.* what variables can we expect to have knowledge of at the time of applying the policy $g$ to intervene on $X$. As noted in Pearl (1994; 2009), the nodes in $\mathbf{Z}_X$ cannot be descendants of $X$ in $G$. Hence, $\mathbf{Z}_X \subseteq \mathbf{V} \setminus \mathrm{De}(X)$. On the other hand, all (proper) ancestors of $X$ are realized before $X$. Since we will be dealing with the case with no latent variables, we can assume that all ancestors of $X$ are observed, and can be used by a policy $g$ to set $X$ to a value $g(\mathbf{Z}_X)$. Thus, we assume[2] that $\mathrm{An}(X) \setminus \{X\} \subseteq \mathbf{Z}_X$. We will then focus on the case where the conditioning set $\mathbf{Z}_X$ is what we call a *observable conditioning set for $X$*, written $\mathbf{Z}_X$, simply meaning that $\mathrm{An}(X) \setminus \{X\} \subseteq \mathbf{Z}_X \subseteq \mathbf{V} \setminus \mathrm{De}(X)$. Finally, we call a map $g \colon R_{\mathbf{Z}_X} \to R_X$ a *policy for $X$*.

**Conditional Causal Bandits**  Recall that a MAB problem consists of an agent pulling an arm $a \in \mathcal{A}$ at each round $t$, resulting in a reward sample $Y_t$ from an unknown distribution associated to the pulled arm (Lattimore and Szepesvári, 2020). We denote the mean reward for arm $a$ by $\mu_a$ and the mean reward for the best arm by $\mu^* = \max_{a \in \mathcal{A}} \mu_a$. The objective can be to maximize the total reward obtained over all the $T$ rounds, or to identify the arm with the highest expected reward (best-arm identification). We can minimize the cumulative regret $\mathrm{Reg_T} = T\mu^* - \sum_{t=1}^{T} \mathbb{E}[Y_t]$ for the former objective, or maximize the probability of selecting the best arm at round $T$ for the latter.

We now introduce a novel type of (causal) MAB problem. Consider the setting where the bandit'S reward is a (endogenous) variable $Y$ in an SCM $\mathfrak{C} = (\mathbf{V}, \mathbf{N}, \mathcal{F}, p_{\mathbf{N}})$, and the arms are the conditional interventions $do(X = g(\mathbf{Z}_X))$, where $X \in \mathbf{V} \setminus \{Y\}$. Furthermore, the agent has knowledge of the causal graph $G$ of $\mathfrak{C}$, but not of the structural assignments $\mathcal{F}$ or the noise distribution $p_{\mathbf{N}}$. We call this a *single-node conditional-intervention causal bandit*, or simply *conditional causal bandit*. The reward distribution for arm $do(X = g(\mathbf{Z}_X))$ is the post-intervention distribution $p_Y^{do(X = g(\mathbf{Z}_X))}$, and is unknown to the agent, since it has no knowledge of $\mathcal{F}$.

Notice that selecting an arm can be subdivided in (i) choosing a node $X$ to intervene on; and (ii) choosing a value to set $X$ to, given the observed variables $\mathbf{Z}_X$. In this paper, we find the minimal set of nodes that need to be considered by the agent in step (i). Note that, since we assume the conditioning sets $\mathbf{Z}_X$ to be predetermined by the intervener and

---

[2] Note that we are not claiming that all variables in $\mathrm{An}(X) \setminus \{X\}$ *need* to be in $\mathbf{Z}_X$ for the best decision to be made, or for our results to hold, but that we *can* always include them in $\mathbf{Z}_X$ under the assumptions of our problem.

the problem context, no choices need to be made in step (ii) regarding which nodes to condition on.

As stressed in Section 1, the novelty of our problem lies in the fact that we deal with *conditional interventions* that are *single-node*. Both of these characteristics of our problem complicate the analysis. Unsurprisingly, searching over conditional interventions is more complicated than over hard or soft interventions. Perhaps more unexpectedly, single-node interventions also make a search for a minimal search space more involved. Indeed, if one allows for interventions on arbitrary sets, one simply needs to intervene on all the parents $\mathrm{Pa}(Y)$ of $Y$. This problem becomes more interesting when unobserved confounding of $Y$ is allowed, in which case simply intervening on $\mathrm{Pa}(Y)$ may not be the optimal approach (Lee and Bareinboim, 2018). Since in our case the agent cannot simply intervene on all the parents of $Y$, the case without unobserved confounding is, as we will see, already complex enough. Extending the results in this paper to cases with unobserved confounding is left as future work (Section 8).

## 3. Conditional-Intervention Superiority

In this section, we will define a preorder $\succeq_Y^c$ of "conditional-intervention superiority" on nodes of an SCM. If $X \succeq_Y^c W$, then $W$ can never be a better node than $X$ to intervene on with a conditional intervention[3]. We will then show that, perhaps surprisingly, this relation is equivalent to another superiority relation, defined in terms of atomic interventions in a deterministic SCM.

**Definition 1** (Conditional-Intervention Superiority). *$X$ is conditional-intervention superior to $W$ relative to $Y$ in $G$, denoted $X \succeq_Y^c W$, if for all SCM with causal graph $G$ there is a policy $g$ for $X$ such that for all policies $h$ for $W$,*

$$\mathbb{E}_{\mathbf{n}} \bar{f}_Y^{do(X = g(\mathbf{Z}_X))}(\mathbf{n}) \geq \mathbb{E}_{\mathbf{n}} \bar{f}_Y^{do(W = h(\mathbf{Z}_W))}(\mathbf{n}), \quad (1)$$

*where $\mathbf{Z}_V$ is a observable conditioning set for $V$. Equivalently, for all $\mathfrak{C}(G)$ one has:*

$$\max_g \mathbb{E}_{\mathbf{n}} \bar{f}_Y^{do(X = g(\mathbf{Z}_X))}(\mathbf{n}) \geq \max_h \mathbb{E}_{\mathbf{n}} \bar{f}_Y^{do(W = h(\mathbf{Z}_W))}(\mathbf{n}). \quad (2)$$

A similar relation can be defined for atomic interventions in deterministic SCMs, where the vector $\mathbf{N}$ of exogenous variables is fixed to a *known* value $\mathbf{n}$ (see Section 2).

**Definition 2** (Deterministic Atomic Intervention Superiority). *Let $X, W, Y$ be nodes of a DAG $G$. $X$ is deterministically atomic-intervention superior to $W$ relative to $Y$, denoted $X \succeq_Y^{\mathrm{det},a} W$, if for every SCM $\mathfrak{C}$ with causal*

---

[3] The relation between nodes introduced by Lee and Bareinboim (2018) is similar, but pertains to multi-node hard interventions.

*graph $G$ and every unit $\mathbf{n}$ there is $X \in R_X$ such that no atomic intervention on $W$ results in a larger $Y$ than the value of $Y$ resulting from setting $X = X$. That is, for all $(\mathfrak{C}, \mathbf{n}) \in \mathfrak{C}(G) \times R_{\mathbf{N}}$:*

$$\exists X \in R_X : \forall w \in R_w, \ \bar{f}_Y^{do(X=X)}(\mathbf{n}) \geq \bar{f}_Y^{do(W=w)}(\mathbf{n}). \tag{3}$$

*Equivalently:*

$$\max_{X \in R_X} \bar{f}_Y^{do(X=X)}(\mathbf{n}) \geq \max_{w \in R_W} \bar{f}_Y^{do(W=w)}(\mathbf{n}). \tag{4}$$

We extend Definitions 1 and 2 for sets of nodes in the obvious way: $\mathbf{X}$ is superior to $\mathbf{W}$ if every node in $\mathbf{W}$ is inferior to some node in $\mathbf{X}$.

**Definition 3.** *Let now $\mathbf{X}, \mathbf{W}$ be sets of nodes of $G$. $\mathbf{X}$ is conditional-intervention superior (respectively deterministic atomic intervention superior) to $\mathbf{W}$, also denoted $\mathbf{X} \succeq_Y^c \mathbf{W}$ (respectively $\mathbf{X} \succeq_Y^{\mathrm{det},a} \mathbf{W}$), if $\forall W \in \mathbf{W}, \exists X \in \mathbf{X}$ such that $X \succeq_Y^c W$ (respectively $X \succeq_Y^{\mathrm{det},a} W$).*

The two relations $\succeq_Y^c, \succeq_Y^{\mathrm{det},a}$ actually coincide (both for nodes and sets of nodes).

**Proposition 4** (Conditional vs Atomic superiority). *Let $X$, $W$, $Y$ be nodes in a DAG $G$. Then $X$ is average conditional-interventionally superior to $W$ relative to $Y$ in $G$ if and only if $X$ is atomic-interventionally superior to $W$ relative to $Y$ in $G$. That is:*

$$X \succeq_Y^c W \iff X \succeq_Y^{\mathrm{det},a} W. \tag{5}$$

Since these two relations are equivalent, we henceforth refer simply to interventional superiority without further specification, and use the symbol $\succeq_Y$ when distinguishing them is not necessary.

*Remark 5.* It is straightfoward to show that both interventional superiority relations are in fact preorders (see Appendix D).

Proposition 4 will simplify our problem. Since deterministic atomic interventions are often easier to reason about, we will use them both in formulating proposals for the minimal search space and in our formal proofs.

*Remark 6.* One may wonder if $\succeq_Y^c$ is also equivalent to the superiority relation for atomic interventions in *non-deterministic* (general) SCMs defined in the natural way: $X \succeq_Y^a W$ iff $\max_X \mathbb{E}_{\mathbf{n}} \bar{f}_Y^{do(X=X)}(\mathbf{n}) \geq \max_w \mathbb{E}_{\mathbf{n}} \bar{f}_Y^{do(W=w)}(\mathbf{n})$. In fact, it is not (see Example 27 in Appendix C).

## 4. Graphical Characterization of the Minimal Globally Interventionally Superior Set

**Goal** Our aim is to develop a method to identify, based on a causal graph $G$, the smallest set of nodes that are "worth

testing" when attempting to maximize $Y$ by performing one atomic intervention. Specifically, regardless of the structural causal model $\mathfrak{C}$ associated with $G$, we want to ensure that the optimal intervention can be discovered within this selected set of nodes. We define this set as follows:

**Definition 7** (GISS and mGISS). *Let $G$ be a DAG with set of nodes $\mathbf{V}$. A globally interventionally superior set (GISS) of $G$ relative to $Y$, is a subset $\mathbf{U}$ of $\mathbf{V} \setminus \{Y\}$ satisfying $\mathbf{U} \succeq_Y (\mathbf{V} \setminus \{Y\}) \setminus \mathbf{U}$. A minimal globally interventionally superior set (mGISS) is a GISS which is minimal with respect to set inclusion.*

This set is unique, so that we can talk of *the* minimal globally interventionally superior set.

**Proposition 8** (Uniqueness of the mGISS). *Let $G$ be a DAG and $Y$ a node of $G$. The minimal globally interventionally superior set of $G$ relative to $Y$ is unique. We denote it by $\mathrm{mGISS}_Y(G)$*

**Intuition** Since the value of $Y$ is completely determined by the values of its parents $A_1, \ldots, A_m$, along with the fixed value $n_Y$ of a noise variable that cannot be intervened upon (see Definition 2), we aim to induce the parents to acquire the combination of values $(a_1^*, \ldots, a_m^*)$ that maximizes $Y$ when $N_Y = n_Y$. If this is not possible to achieve using a single intervention, we aim to obtain the best combination possible. Clearly, the parents of $Y$ themselves need to tested by bandit algorithms: there may be one parent on which $Y$ is highly dependent, in such a way that there is a value of that parent which will maximize $Y$. In the particular case where $Y$ has a single parent $A$, that node is the only node worth intervening on, since all other nodes can only influence $Y$ through $A$. Indeed, if $a^* \in R_A$ is the value of $A$ which maximizes $Y$, it is not necessary to try to find an intervention on ancestors of $A$ which results in $A = a^*$: just set $A = a^*$ directly (Figure 1c). If $Y$ has two or more parents, it is possible that a single intervention on one of the $A_i$ does not yield the best possible outcome. Instead, a better configuration (potentially even the ideal case $(a_1^*, \ldots, a_m^*)$) may be achieved by intervening *on a common ancestor* of some or all of the $A_i$ (Figure 1a). Notice that $X_0$ is also a common ancestor of $A_1, A_2$, but one is never better off intervening on $X_0$ than on $X_1$. This seems to indicate that testing interventions on, for instance, all lowest common ancestors (LCAs, see Appendix A) of the parents of $Y$, and only them, is necessary. While this works in Figure 1a, it fails for a graph such as Figure 1d, where $X$ needs to be tested and yet it is not in $\mathrm{LCA}(A_1, A_2) = \{A_1\}$. This suggests that we need to define a stricter notion of common ancestor to make progress in characterizing $\mathrm{mGISS}_Y(G)$.

**Definition 9** (Lowest Strict Common Ancestors of a Pair of Nodes). *The node $V \in \mathbf{V}$ is a strict common ancestor of $X, Y \in \mathbf{V}$ if $V$ is a common ancestor of $X, Y$ from which both $X$ and $Y$ can be reached from $V$ with paths $V \dashrightarrow X$*

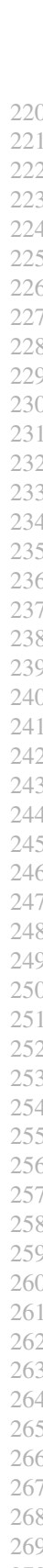

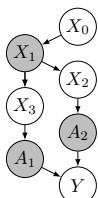

(a) Two parents with a lowest common ancestor. It may happen that setting $X_1$ a certain value will set $(A_1, A_2)$ to $(a_1^*, a_2^*)$, while intervening on one of the $A_i$ would not.

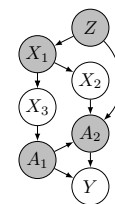

(b) The heuristics justifying the need to test the LSCA $X_1$ of the parents $A_1, A_2$ of $Y$ can be repeated for $X_1$ and $A_2$. Thus, $Z$ should be tested as well.

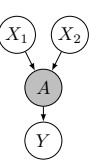

(c) Single parent. Setting $A$ to $a^*$ is the best option.

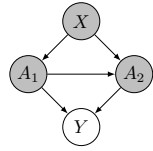

(d) Just as in Figure 1a, $X$ may need to be intervened upon. However, $\mathrm{LCA}(A_1, A_2) = \{A_1\} \not\ni X$.

Figure 1: Examples illustrating heuristics behind the graphical characterization of the minimal interventionally superior set. The gray nodes are those that should be tested by conditional causal bandit algorithms.

*and $V \dashrightarrow Y$ not containing $Y$ and $X$, respectively. The set of strict common ancestors of $X, Y$ is denoted $\mathrm{SCA}(X, Y)$. Furthermore, $V$ is a lowest strict common ancestor of $X, Y \in \mathbf{V}$ if $V$ is a minimal element of $\mathrm{SCA}(X, Y)$ with respect to the ancestor partial order $\preccurlyeq$. The set of lowest strict common ancestors of $X, Y$ is denoted $\mathrm{LSCA}(X, Y)$.*

**Definition 10** (Lowest Strict Common Ancestors of a Set). *Let $\mathbf{U} \subseteq \mathbf{V}$ and $V \in \mathbf{V} \setminus \mathbf{U}$. The node $V$ is a lowest strict common ancestor of $\mathbf{U}$ if it is a lowest strict common ancestor of some pair of nodes $U, U'$ in $\mathbf{U}$. The set of lowest strict common ancestors is denoted $\mathrm{LSCA}(\mathbf{U})$. That is,*

$$\mathrm{LSCA}(\mathbf{U}) := \{V \in \mathbf{V} \setminus \mathbf{U} : \exists U, U' \in \mathbf{U} \\ s.t.\ V \in \mathrm{LSCA}(U, U')\}. \tag{6}$$

Our heuristic argument so far suggests that we need to test the parents of $Y$ and their LSCAs. However, there are additional nodes that must be considered: the reasoning for testing the lowest strict common ancestors of the parents can be repeated. For instance, in Figure 1b, the best possible configuration of the $A_i$ may be achieved by intervening on $Z$. Such an intervention could result in a combination of values of $X_1$ and $A_2$ that leads to the best possible combinations of $A_1$ and $A_2$. This suggests that the $\mathrm{mGISS}_Y(G)$ should be determined by recursively finding all the LSCAs of the parents of $Y$, then the LSCAs of that set, and so on, ultimately resulting in what we call the "LSCA closure of

the parents of $Y$", denoted $\mathcal{L}^\infty(\mathrm{Pa}(Y))$. In the remainder of this section, we formally define $\mathcal{L}^\infty(\mathrm{Pa}(Y))$, find a simple graphical characterization for it, and prove that it indeed equals $\mathrm{mGISS}_Y(G)$.

**Definition 11** (LSCA closure). *For every $i \in \mathbb{N}$ we define the $i^{th}$-order LSCA set $\mathcal{L}^i(\mathbf{U})$ of $\mathbf{U} \subseteq \mathbf{V}$ as follows:*

$$\mathcal{L}^0(\mathbf{U}) := \mathbf{U} \\ \mathcal{L}^i(\mathbf{U}) := \mathrm{LSCA}(\mathcal{L}^{i-1}(\mathbf{U})) \cup \mathcal{L}^{i-1}(\mathbf{U}). \tag{7}$$

*The LSCA closure $\mathcal{L}^\infty(\mathbf{U})$ of $\mathbf{U}$ is given by*

$$\mathcal{L}^\infty(\mathbf{U}) := \mathcal{L}^{k^*}(\mathbf{U}), \\ where\ \ k^* = \min\{i \in \mathbb{N} : \mathcal{L}^i(\mathbf{U}) = \mathcal{L}^{i+1}(\mathbf{U})\}. \tag{8}$$

*Remark* 12. Notice that the existence of the $k^*$ in Equation (8) is guaranteed, since by construction $\mathcal{L}^{i+1}(\mathbf{U}) \subseteq \mathcal{L}^i(\mathbf{U}) \subseteq \mathbf{V}$ for all $i \in \mathbb{N}$ and $\mathbf{V}$ is finite.

*Example* 13. Consider the graph in Figure 1b and set $\mathbf{U} = \{A_1, A_2\}$. Then, $\mathcal{L}^0(\mathbf{U}) = \{A_1, A_2\}, \mathcal{L}^1(\mathbf{U}) = \{X_1, A_1, A_2\}, \mathcal{L}^2(\mathbf{U}) = \{Z, X_1, A_1, A_2\} = \mathcal{L}^3(\mathbf{U})$. Hence, $\mathcal{L}^\infty(\mathbf{U}) = \{Z, X_1, A_1, A_2\}$.

We will introduce the notion of "$\Lambda$-structures" (Figure 2), which provides an alternative, elegant, simple graphical characterization of $\mathcal{L}^\infty(\mathrm{Pa}(Y))$. It will also be instrumental in the proofs of the main results of this paper.

**Definition 14** ($\Lambda$-structure). *Let $V, A, B \in \mathbf{V}$. Furthermore, let $\pi_A : V \dashrightarrow A$, $\pi_B : V \dashrightarrow B$ be paths. The tuple $(V, \pi_A, \pi_B)$ is a $\Lambda$-structure over $(A, B)$ if $\pi_A$ and $\pi_B$ only intersect at $V$. Now, let $\mathbf{U}, \mathbf{W} \subseteq \mathbf{V}$. The node $V$ is said to form a $\Lambda$-structure over $(\mathbf{U}, \mathbf{W})$ if there are nodes $U \in \mathbf{U}$ and $W \in \mathbf{W}$, and paths $\pi_U : V \dashrightarrow U$, $\pi_W : V \dashrightarrow W$ such that $(V, \pi_U, \pi_W)$ is a $\Lambda$-structure over $(U, W)$. Denote by $\Lambda(\mathbf{U}, \mathbf{W})$ the set of all nodes forming a $\Lambda$-structure over $(\mathbf{U}, \mathbf{W})$.*

Notice that, if $V \in \mathbf{U} \cap \mathbf{W}$, then trivially $V \in \Lambda(\mathbf{U}, \mathbf{W})$: just take the trivial paths $\pi = \pi' = (V)$.

**Theorem 15** (Simple Graphical Characterization of LSCA Closure). *A node $V \in \mathbf{V}$ is in the LSCA closure $\mathcal{L}^\infty(\mathbf{U})$ of $\mathbf{U} \subseteq \mathbf{V}$ if and only if $V$ forms a $\Lambda$-structure over $(\mathbf{U}, \mathbf{U})$. I.e. $\mathcal{L}^\infty(\mathbf{U}) = \Lambda(\mathbf{U}, \mathbf{U})$.*

We are now ready for the main result of this paper: that the LSCA closure $\mathcal{L}^\infty(\mathrm{Pa}(Y))$ of the parents of $Y$ is the minimimal globally interventionally superior set with respect to $Y$.

**Theorem 16** (Superiority of the LSCA Closure). *Let $G$ be a causal graph and $Y$ a node of $G$ with at least one parent. Then, the LSCA closure $\mathcal{L}^\infty(\mathrm{Pa}(Y))$ of the parents of $Y$ is the minimal globally interventionally superior set $\mathrm{mGISS}(G)$ of $G$ relative to $Y$.*

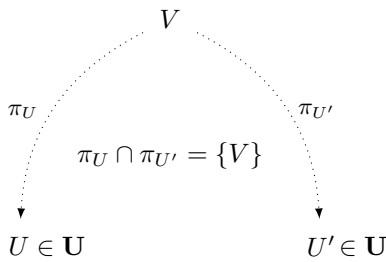

Figure 2: A $\Lambda$-structure over $(\mathbf{U}, \mathbf{U})$. The LSCA closure $\mathcal{L}^\infty(\mathbf{U})$ of a set $\mathbf{U}$ is the set of all such structures.

We emphasize that, due to Proposition 4, this graphical characterization of the $\mathrm{mGISS}_Y(G)$ is valid both for conditional interventions in a probabilistic causal model as for atomic interventions in a deterministic causal model (*i.e.* a causal model with known $\mathbf{n}$).

## 5. Algorithm to Find the Minimal Globally Interventionally Superior Set

---

**Algorithm 1** C4

---
1: **input:** DAG $G = (\mathbf{V}, E)$, set of nodes $\mathbf{U} \subseteq \mathbf{V}$
2: **output:** The closure $\mathcal{L}^\infty(\mathbf{U})$
3: $S \leftarrow \mathbf{U}$                  $\triangleright$ initialize closure
4: $\mathfrak{c}[V] \leftarrow V$ for $V \in \mathbf{U}$      $\triangleright$ initalize connectors
5: $\mathfrak{c}[V] \leftarrow \mathrm{NULL}$ for $V \in \mathbf{V} \backslash \mathbf{U}$    $\triangleright$ initalize connectors
6: **for** $V \in \mathbf{V} \backslash \mathbf{U}$ in reverse topological order **do**
7:     $C \leftarrow \{\mathfrak{c}[V'] : V' \in \mathrm{Ch}(V), \mathfrak{c}[V'] \neq \mathrm{NULL}\}$
8:     **if** $|C| = 1$ **then**
9:        $\mathfrak{c}[V] \leftarrow X$ where $C = \{X\}$
10:    **else if** $|C| > 1$ **then**
11:       $\mathfrak{c}[V] \leftarrow V, S \leftarrow S \cup \{V\} \triangleright V$ is added to closure
12: **return** $S$

---

The **C**losure **C**omputation via **C**hildren with Multiple **C**onnectors (C4) Algorithm (Algorithm 1) computes the closure $\mathcal{L}^\infty(\mathbf{U})$ in $O(|\mathbf{V}| + |E|)$ time, using *connectors*:

**Definition 17** (Connector). *Let $G = (\mathbf{V}, E)$ be a DAG, $\mathbf{U} \subseteq \mathbf{V}$, $V \in \mathbf{V}$. A node $X \in \mathbf{V}$ is a $\mathbf{U}$-connector of $V$ (in $G$) iff $X$ is a maximal element of $De(V) \cap \mathcal{L}^\infty(\mathbf{U})$ with respect to the ancestor partial order $\preccurlyeq$.*

Note that $V \in \mathcal{L}^\infty(\mathbf{U})$ iff $V$ is its own connector. A connector $X$ can be gotten to from $V$ only via paths excluding $\mathcal{L}^\infty(\mathbf{U}) \backslash \{X\}$. Lemma 18 shows that in fact the existence of one such path is sufficient (and necessary) for a node to be a connector; furthermore, it establishes that a connector—if it exists—is unique, and so we call it *the* connector. See Figure 3 for an example.

**Lemma 18** (Uniqueness and Characterization of Connectors). *Let $G = (\mathbf{V}, E)$ be a DAG, $\mathbf{U} \subseteq \mathbf{V}$, $V \in \mathbf{V}$. If $V$*

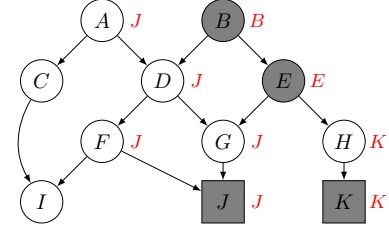

Figure 3: Illustration of the connectors in a graph. The square nodes belong to $\mathbf{U}$, the connector of each node is written in red next to its node, and the LSCA closure $\mathcal{L}^\infty(\mathbf{U})$ consists of the gray nodes.

*has a $\mathbf{U}$-connector $V'$, then $V'$ is the unique node for which there is a path $\pi_{V'} = V \dashrightarrow V'$ s.t. $\pi_{V'} \cap \mathcal{L}^\infty(\mathbf{U}) = \{V'\}$.*[4]

Let us informally sketch the idea behind C4. By Lemma 18, it follows that the connector of $V$ is the unique node from $\mathcal{L}^\infty(\mathbf{U})$ included in all paths from $V$ to $\mathcal{L}^\infty(\mathbf{U})$. Assume $V \notin \mathbf{U}$. Then $V$ has access to $\mathbf{U}$ only via non-trivial paths, and the second node in each such path is a child of $V$. Therefore, every path from $V$ to $\mathbf{U}$ must go through a node from the set $C$ of $V$'s children's connectors. If $C = \emptyset$, $V$ has no path to $\mathbf{U}$, so $V \notin \Lambda(\mathbf{U}, \mathbf{U}) = \mathcal{L}^\infty(\mathbf{U})$. Furthermore, $V$ has no path to $\mathcal{L}^\infty(\mathbf{U})$, so it has no connector. If $C = \{X\}$, then all paths from $V$ to $\mathbf{U}$ must coincide at $X$, and again $V \notin \Lambda(\mathbf{U}, \mathbf{U}) = \mathcal{L}^\infty(\mathbf{U})$. Moreover, since $V \notin \mathcal{L}^\infty(\mathbf{U})$, every path from $V$ to $\mathcal{L}^\infty(\mathbf{U})$ must go through $X$, so $X$ is $V$'s connector. If $|C| > 1$, then one can show that there is a $\Lambda$-structure from $V$ to a pair of its children's connectors, and as these connectors are in $\mathcal{L}^\infty(\mathbf{U})$, Theorem 15 implies $V \in \mathcal{L}^\infty(\mathcal{L}^\infty(\mathbf{U}))$. However, it is easily seen that $\mathcal{L}^\infty(\mathcal{L}^\infty(\mathbf{U})) = \mathcal{L}^\infty(\mathbf{U})$, so $V \in \mathcal{L}^\infty(\mathbf{U})$ and is its own connector. Accordingly, C4 sets $v \in \mathcal{L}^\infty(U)$ iff $|C| > 1$. Theorem 19 formalizes our intuition, and Theorem 20 establishes linear running time.

**Theorem 19.** *C4 correctly computes $\mathcal{L}^\infty(U)$.*

**Theorem 20.** *C4 runs in $O(|\mathbf{V}| + |E|)$ time.*

In our experiments (Section 6), we accelerate C4 by pre-empting the computation of $C$ as soon as two members of $C$ are found, as this ensures $|C| > 1$.

## 6. Experimental Results

We evaluate $C_4$ on both random and real graphs. Additionally, we examine the impact of our method on the cumulative regret of a bandit algorithm.

**Search Space Reduction in Random Graphs** We applied the C4 algorithm to randomly generated DAGs using

---

[4]If $V$ is its own connector, the path is trivial.

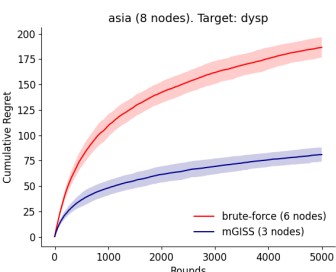 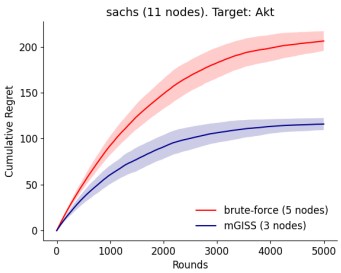 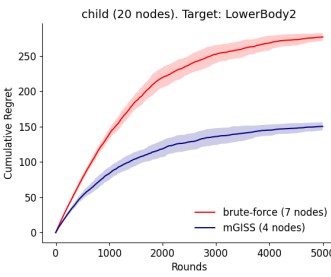

Figure 4: Comparison of cumulative regret curves for node selection using a UCB-based bandit algorithm for conditional interventions, with (mGISS) and without (brute-force) pruning the search space. These curves were obtained by averaging over 500 runs, on four `bnlearn` datasets (`asia`, `sachs`, `child`). For every dataset, pruning the search space with the C4 algorithm results in faster convergence and smaller values of regret.

the Erdős-Rényi model for $N$ graphs and probability $p$ (Erdős and Rényi, 1959) adapted to DAG-generation[5]. We generated 1000 graphs using $20, 100, 300$, and $500$ nodes, and varying the expected (total) degree of nodes from 2 to 11 in steps of 3. For each graph $G$, we set the target $Y$ to be the node with the most ancestors, used C4 to compute $\mathcal{L}^\infty(\mathrm{Pa}(Y)) = \mathrm{mGISS}_Y(G)$, and calculated the fraction of nodes in $\mathrm{An}(Y) \setminus \{Y\}$ that remain in $\mathrm{mGISS}_Y(G)$. The results revealed that, for a given number of nodes, graphs with lower expected degrees benefit more from our method (*i.e.* their $\mathrm{mGISS}_Y(G)$ correspond to smaller fractions of $\mathrm{An}(Y) \setminus \{Y\}$). Furthermore, for a fixed expected degree, our method is more effective for higher numbers of nodes. For example, for graphs with $500$ nodes, the mGISS retained, on average, $17\%, 29\%, 62\%$ and $77\%$ of the nodes, for expected degrees of $2, 5, 8$ and $11$, respectively. Moreover, graphs with an expected degree of $5$ saw these numbers decrease from $70\%$ at 20 nodes to $47\%, 35\%$ and $29\%$ for $100, 300$ and $500$ nodes, respectively. The complete results are presented in histograms in Figure 5 (Appendix G). These results are not surprising: if the average degree is small compared to the number of nodes, the edge density is small, in which case we expect fewer $\Lambda$-structures to form over $\mathrm{Pa(Y)}$.

Graphs modeling real-world systems tend to have low average degrees, as can be seen in the graphs from the popular Bayesian network repository `bnlearn`. Therefore, we expect our method to be especially effective in those graphs. We test this below.

**Search Space Reduction in Real-World Graphs**   We tested our method in most graphs from the `bnlearn` repository[6], as well as on a graph representing the causal relationships between train delays in a segment of the railway

system of the Netherlands (see Appendix G). For each graph, we set $Y$ to be the node with most ancestors[7]. The results are presented in the bar plot of Figure 6 (Appendix G). This confirmed that realistic models with larger graphs tended to benefit more from our method, with a reduction of over $90\%$ of the search space for some of the largest models. Notice also that these models indeed have relatively small average degrees, all below 4.0. From this, we conclude that we can expect our method to be useful when reducing the search space of conditional causal bandit tasks in real-world causal models, especially when they are large.

**Impact on Conditional Intervention Bandits**   We present empirical evidence that restricting the node search space to the mGISS allows a straightforward UCB-based[8] algorithm (which we call CondIntUCB) for conditional causal bandits to converge more rapidly to better nodes. As explained in Section 2, on each round the algorithm must (i) choose which node $X$ to intervene on; and (ii) choose the value for $X$, given its conditioning set $Z_X$[9]. Choice (i) employs UCB over nodes, while choice (ii) utilizes a UCB instance specific to the conditioning set value. In other words, for each realization of $\mathbf{Z}_X$ (each context) there is a UCB. This is identical to what is described in Lattimore and Szepesvári (2020, §18.1) for contextual bandits with one bandit per context. The cumulative regret[10] is computed with respect to node choice, since we want to see how our node selection method affects the quality of node choice by CondIntUCB. We use 3 real-world datasets from the `bnlearn` repository, and again choose the node of each dataset with the most an-

---

[5]After fixing a total order $\trianglelefteq$ on the nodes, each pair of nodes $V$, $u$ with $V \trianglelefteq u$ is assigned an edge $(V, u)$ with probability $p$. The value $p$ can be used to control the expected degree.

[6]All that can be imported in Python using the library `pgmpy`.

[7]We also require $Y$ to have more than one parent, to avoid the trivial case with $|\mathrm{mGISS}_Y(G)| = 1$.

[8]The Upper Confidence Bound (UCB) algorithm is a widely used MAB algorithm. See *e.g.* Lattimore and Szepesvári (2020).

[9]For simplicity, since the smallest possible observable conditioning set is $\mathrm{An}(X)$ (see Section 2), we use $\mathbf{Z}_X = \mathrm{An}(X)$.

[10]For the computation of regret, we use the estimated best arm, defined as the arm that most runs concluded to be the best at the end of training.

cestors as the target[7]. These datasets were selected because their graphical structures are non-trivial[11] and both $\mathrm{An}(Y)$ and $\mathrm{mGISS}_Y(G)$ are sufficiently small to allow experimentation with our setup. For each dataset, we run CondIntUCB 500 times and plot the two average cumulative regret curves along with their standard deviations, corresponding to using all nodes (brute-force) and the mGISS nodes (Figure 4). The total number of rounds is chosen as to observe (near) convergence. These results show that cumulative regret curves can be significantly improved—meaning that better nodes are selected earlier for applying conditional interventions—if the search space over nodes is pruned using our C4 algorithm.

## 7. Related Work

Recent research has explored the integration of causality and multi-armed bandit (MAB) frameworks. As mentioned in Section 1, Lattimore et al. (2016) introduced the original causal bandit problems, which involve hard interventions in causal models. Subsequent works (Sen et al., 2017; Yabe et al., 2018; Lu et al., 2020; Nair et al., 2021; Sawarni et al., 2023; Maiti et al., 2022; Feng and Chen, 2023) proposed algorithms for variants of causal bandits with both hard and soft interventions, budget constraints, and unobserved confounders, all under specific assumptions, such as binary variables, simple graphs, or known post-intervention distributions. Note that we do not make such assumptions.

Recent works in "contextual causal bandits" address interventions that account for context, bearing a superficial resemblance to our problem. However, our problem remains distinct. In Madhavan et al. (2024), the term "contexts" is used in a very different way, actually referring to different graphs as opposed to different variable values. Subramanian and Ravindran (2022; 2024) tackle the scenario in which an intervention is performed, with knowledge of a given set of context variables, on a *pre-chosen* variable $X$ that has an edge into $Y$ (and no other outgoing edges). This approach can be understood as selecting a conditional intervention for a predefined node from a very simple graph. In contrast, in our setting we need to choose what variable to intervene on to begin with, and there are no restrictions on the causal graph.

All of the works described above proposed algorithms which aim at accelerating learning by utilizing knowledge of the causal model. As explained in Section 1, this contrasts with the work by Lee and Bareinboim (2018; 2019), which, just like our work, uses knowledge of the causal graph to find a minimal search space (over the nodes) for causal bandits. While they focus on multi-node, hard interventions, we focus on single-node, conditional interventions.

---

[11]In contrast, the `cancer` dataset, for example, only has nodes whose mGISS is either all of the node's ancestors or a single node.

The work of Lee and Bareinboim (2020) presents an interesting connection to our work. Given a causal graph, they study the sets of pairs $(\mathrm{node}, \mathrm{context}(\mathrm{node}))$ (referred to as "scopes") that may correspond to an optimal (multi-node) intervention policy where each node $X$ in a scope is intervened on according to a policy $\pi_X(X \mid \mathrm{context}(X))$. This is a challenging problem, and they do not provide a full characterization of these optimal scopes, instead deriving a set of rules that can be used to compare certain pairs of scopes. In this paper, we instead assume that the practitioner knows the appropriate conditioning set $\mathbf{Z}_X$ (context) to use and impose only minimal restrictions on what $\mathbf{Z}_X$ can be, focusing instead on choosing the nodes that can yield the best results. While Lee and Bareinboim (2020) consider multi-node interventions, it would be interesting in future work to adapt their ideas to the single-node case to identify the smallest $\mathbf{Z}_X$ sets for which the best policy can still be found. Such an approach could further accelerate learning by MAB algorithms.

## 8. Conclusion

In this paper, we introduced the conditional causal bandit problem, where the agent only has knowledge of the causal graph $G$, the arms are conditional interventions, and the reward variable belongs to $G$. The theoretical contributions include a rigorous, simple graphical characterization of the minimal set of nodes which is guaranteed to contain the node with the optimal conditional intervention, and the C4 algorithm, which computes this set in linear time. Empirical results validate that our approach substantially prunes the search space in both real-world and sparse randomly-generated graphs. Furthermore, integrating mGISS with a UCB-based conditional bandits algorithm showcased improved cumulative regret curves.

As mentioned in Section 7, a possible future research direction is to identify the smallest conditioning set(s) $\mathbf{Z}_X$, rather than assuming, as we do, that the practitioner or problem setting determines them. Another relevant direction for future work is the incorporation of latent variables. On the practical side, instead of combining C4 with the simple CondIntUCB, one could replace CondIntUCB with any other conditional bandit algorithm that leverages the model's causal structure. As discussed in Section 7, no such algorithm currently exists. Nevertheless, we expect that combining C4 with any future algorithm for causal bandits with conditional interventions will be advantageous, as it reduces the number of arms that need to be considered.

## Impact Statement

Our work proposes tools to enhance the efficiency of AI agents in decision-making problems. Solutions to these

kinds of problems lead to well-established issues, particularly when applied blindly — that is, when the algorithms' conclusions are used to make real-world decisions without assessing potential dangers or ethical concerns not captured by the mathematical model. Mitigating such risks may require, for example, the regulation of these tools and the education of users regarding their limitations.

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

## A. Directed Acyclic Graphs

All graphs in this paper are directed acyclic graphs (DAGs). Every path is assumed to be directed. A path $\pi$ in a graph $G = (\mathbf{V}, E)$ is a tuple of nodes such that each node $X$ in the path has an outgoing arrow from $X$ to the next node in the tuple[12]. For $X \in \mathbf{V}$, we denote by $\mathrm{Pa}(X)$, $\mathrm{Ch}(X)$, $\mathrm{De}(X)$ and $\mathrm{An}(X)$ the sets of parents, children, descendants and ancestors of $X$, respectively. We denote by $\pi\colon X \dashrightarrow Y$ a path starting at node $X$ and ending at node $Y$, and $\mathring{\pi}$ denotes the path formed by the inner nodes of $\pi$. By abuse of notation, we often perform set operations such as $\pi_1 \cap \pi_2$ between paths, which implicitly means that these operations are performed on the sets of nodes belonging to the paths. Tuples with a single node are also considered to be paths, and are said to be *trivial*. Also, if $B \in \pi\colon X \dashrightarrow Y$, then the paths $\pi|^Z\colon Z \dashrightarrow Y$ and $\pi|_Z\colon X \dashrightarrow Z$ are the paths resulting from removing from $\pi$ all nodes before and after $Z$, respectively. Every node is an ancestor of itself, so that the relation $\preccurlyeq$ defined by $X \preccurlyeq Y \iff Y \in \mathrm{An}(X)$ is a partial order. Given a set $\mathbf{U}$ of nodes, we denote by $\max_{\preccurlyeq}[\mathbf{U}]$ the set of maximal elements of $\mathbf{U}$ with respect to $\preccurlyeq$. We call this the *ancestor partial order*. If there is a non-trivial path from $X$ to $Y$, then $Y$ is said to be *reachable* from $X$. The set of common ancestors of nodes $X$ and $Y$ is denoted $\mathrm{CA}(X, Y) = \mathrm{An}(X) \cap \mathrm{An}(Y) = \{Z \in \mathbf{V} : Z \preccurlyeq X \wedge Z \preccurlyeq Y\}$. Finally, the *degree* of a node in a DAG is the sum of the incoming and outgoing arrows of that node.

We also make use of a lesser-known graph theory concept, relevant for this paper: the "lowest common ancestors" of nodes $(X, Y)$. These are common ancestors that don't reach any other common ancestors, intuitively making them the "closest" to $(X, Y)$.

**Definition 21** (Lowest Common Ancestors in a DAG (Bender et al., 2005)). *Let $X, Y$ be nodes of a DAG $G = (\mathbf{V}, E)$. A lowest common ancestor (LCA) of $X$ and $Y$ is a maximal element of $\mathrm{CA}(X, Y)$ with respect to the ancestor partial order $\preccurlyeq$. The set of all lowest common ancestors of $X$ and $Y$ is denoted $\mathrm{LCA}(X, Y)$.*

For example, in Figure 1a, $\mathrm{LCA}(A_1, A_2) = \{X_1\}$, whereas in Figure 1b, $\mathrm{LCA}(A_1, A_2) = \{A_1\}$.

## B. Unrolled Assignments

The structural assignments of an SCM can be utilized to express any endogenous variable as a function of the exogenous variables only. This is achieved by composing the assignments until reaching the exogenous variables. Our proofs will rely on these functions, which we will refer to as "unrolled assignments", since we "unroll" the expressions for the endogenous variables until only exogenous variables are left. We define them formally by induction as follows:

**Definition 22** (Unrolled Assignment). *We define the* unrolled assignment $\bar{f}_X\colon R_{\mathbf{N}} \to R_X$ *of any (exogenous or endogenous) variable $X$ from an SCM $\mathfrak{C} = (\mathbf{V}, \mathbf{N}, \mathcal{F}, p_{\mathbf{N}})$ by induction. For $X = N_i \in \mathbf{N}$, define $\bar{f}_X(\mathbf{n}) := n_i$. Now, let $\unlhd$ be a topological order on $G$ where the first elements are the endogenous variables with no endogenous parents. Let $S$ be the poset $(\mathbf{V}, \unlhd)$. In ascending order, take $X \in S$, and define:*

$$\bar{f}_X(\mathbf{n}) := \begin{cases} f_X(n_X), & \text{if } \mathrm{Pa}(X) = \emptyset \\ f_X(\bar{f}_{\mathrm{Pa}(X)}(\mathbf{n}), n_X), & \text{otherwise} \end{cases}, \tag{9}$$

*where $\bar{f}_{\mathrm{Pa}(X)}(\mathbf{n}) = (\bar{f}_{\mathrm{Pa}(X)_1}(\mathbf{n}), \ldots, \bar{f}_{\mathrm{Pa}(X)_{m_X}}(\mathbf{n}))$ and $m_X = |\mathrm{Pa}(X)|$.*

Additionally, we can consider $X$ as a function of both exogenous variables and a chosen endogenous variable $B$. To achieve this, we substitute the assignments until we reach either $B$ or the exogenous variables, thereby "unrolling" the dependencies until we reach the exogenous variables or we are blocked by $B$.

**Definition 23** (Blocked Unrolled Assignment). *Let $X, B$ endogenous variables from an SCM $\mathfrak{C} = (\mathbf{V}, \mathbf{N}, \mathcal{F}, p_{\mathbf{N}})$ We define the* unrolled assignment $\bar{f}_X[B]\colon R_B \times R_{\mathbf{N}} \to R_X$ *of $X$ blocked by $B$ by induction. Let $S$ be the poset from Definition 22. In ascending order, take $X \in S$, and define:*

$$\bar{f}_X[B](B, \mathbf{n}) := \begin{cases} \bar{f}_X(\mathbf{n}), & \text{if } X \notin \mathrm{De}(B) \\ B, & \text{if } X = B \\ f_X(\bar{f}_{\mathrm{Pa}(X)}[B](B, \mathbf{n}), n_X) & \text{otherwise} \end{cases}, \tag{10}$$

*where $\bar{f}_{\mathrm{Pa}(X)}[B](\mathbf{n}) = (\bar{f}_{\mathrm{Pa}(X)_1}[B](B, \mathbf{n}), \ldots, \bar{f}_{\mathrm{Pa}(X)_{m_X}}[B](B, \mathbf{n}))$ and $m_X = |\mathrm{Pa}(X)|$.*

---

[12]Since all DAGs we are considering in this paper come from SCMs, there is at most one arrow between any two nodes, so that a tuple of nodes is enough to define a path. For a general graph one would have to specify a list of edges.

*Remark* 24. Strictly speaking, $\bar{f}_X$ is not a function of all the values of all the noise variables, but only of the exogenous variables $N_W$ associated with endogenous variables $W$ that $Y$ depends on. Similarly, $\bar{f}_X[B]$ is also not a function of all the values of all the noise variables. Namely, if $X$ only depends on an endogenous variable $W$ through $B$, then $n_W$ will never appear in the expression for $\bar{f}_X[B]$, and the same holds in case $B = W$. A more accurate notation would reflect these facts, writing the unrolled assignments as functions of the specific noise variables that can affect them, rather than as functions of all noise variables. We opted not to adopt this notation to avoid complicating the notation and conceptual simplicity of these quantities.

# C. Conditional Superiority vs Deterministic Atomic Superiority

We will show that conditional intervention superiority is equivalent to deterministic atomic intervention superiority. This result will help prove results about the former by making use of the former, which is mathematically simpler and easier to reason about.

*Notation.* We denote by $G^*$ the graph resulting from adding to a causal graph $G$ the exogenous variables as nodes, and an edge $N_{X_i} \to X_i$ for each exogenous variable $N_{X_i}$.

**Lemma 25** (Conditional Intervention vs Atomic Intervention). *Let $X, Y$ be endogenous variable of $\mathfrak{C}$ and Let $A$ be a set of endogenous variables of an SCM $\mathfrak{C}$, and. When evaluated at a setting $\mathbf{n}$, the unrolled assignment of $Y$ after a conditional intervention $do(X = g(A))$ coincides with the unrolled assignment of $Y$ after the atomic intervention $do(X = \bar{f}_A(\mathbf{n}))$. That is:*

$$\bar{f}_Y^{do(X=g(A))}(\mathbf{n}) = \bar{f}_Y^{do(X=g(\bar{f}_A(\mathbf{n})))}(\mathbf{n}).$$

*Proof.* This result can be proved by induction in a similar way to Lemma 36.

Let $X$ be an endogenous variable. We want to prove that the expression holds for any variable $Y$. We will prove this by induction on a topological order $\trianglelefteq$ on the nodes of $G^*$ such that the first elements are precisely the exogenous variables, *i.e.* $N \trianglelefteq Z$ whenever $N \in \mathbf{N}$ and $Z \in \mathbf{V}$.

The result is true for the exogenous variables. Indeed, for $Y \in \mathbf{N}$, and making use of Lemma 36, we have that $\bar{f}_Y^{do(X=g(\bar{f}_A(\mathbf{n})))}(\mathbf{n}) = \bar{f}_Y[X](g(\bar{f}_A(\mathbf{n})), \mathbf{n}) = \bar{f}_Y(\mathbf{n}) = Y = \bar{f}_Y^{do(X=g(A))}(\mathbf{n})$, since $Y \notin \mathrm{De}(X) \cup \{X\}$ and $Y$ is exogenous (both in the pre- and post-intervention (both conditional and atomic) structural causal models). This establishes the base case of the induction.

Now let $Y$ be endogenous. For the inductive step, we will prove that, if the result is true for the parents $\mathrm{Pa}_{G^*}(Y)$ of $Y$ in $G^*$ (induction hypothesis), then it is also true for $Y$. Assume the antecedent (induction hypothesis). There are three possibilities: $Y \in \mathrm{De}(X) \setminus \{X\}, Y = X$ or $Y \notin \mathrm{De}(X)$. In case $Y \in \mathrm{De}(X) \setminus \{X\}$:

$$
\begin{aligned}
\bar{f}_Y^{do(X=g(\bar{f}_A(\mathbf{n})))}(\mathbf{n}) &\overset{\text{def}}{=} f_Y^{do(X=g(\bar{f}_A(\mathbf{n})))}(\bar{f}_{\mathrm{Pa}(Y)}^{do(X=g(\bar{f}_A(\mathbf{n})))}(\mathbf{n}), n_Y) \\
&= f_Y(\bar{f}_{\mathrm{Pa}(Y)}^{do(X=g(\bar{f}_A(\mathbf{n})))}(\mathbf{n}), n_Y) \\
&\overset{\text{I.H.}}{=} f_Y(\bar{f}_{\mathrm{Pa}(Y)}^{do(X=g(A))}(\mathbf{n}), n_Y) \\
&= f_Y^{do(X=g(A))}(\bar{f}_{\mathrm{Pa}(Y)}^{do(X=g(A))}(\mathbf{n}), n_Y) \\
&\overset{\text{def}}{=} \bar{f}_Y^{do(X=g(A))}(\mathbf{n}),
\end{aligned}
\tag{11}
$$

where in the second and fourth equalities we used that $f_Y^{do(X=g(\bar{f}_A(\mathbf{n})))} = f_Y = f_Y^{do(X=g(A))}$. We also used that $Pa(Y)$ is unchanged by these interventions. If instead $Y = X$, then one has:

$$
\begin{aligned}
\bar{f}_X^{do(X=g(A))}(\mathbf{n}) &\overset{\text{def}}{=} f_X^{do(X=g(A))}(\bar{f}_{\mathrm{Pa}_{G^{do(X=g(A))}}(X)}^{do(X=g(A))}(\mathbf{n}), n_X) \\
&= f_X^{do(X=g(A))}(\bar{f}_A^{do(X=g(A))}(\mathbf{n}), n_X) \\
&= g(\bar{f}_A(\mathbf{n}), n_X),
\end{aligned}
\tag{12}
$$

and also:

$$
\begin{aligned}
\bar{f}_X^{do(X=g(\bar{f}_A(\mathbf{n})))}(\mathbf{n}) &\overset{\text{def}}{=} f_X^{do(X=g(\bar{f}_A(\mathbf{n})))}(\bar{f}_{\mathrm{Pa}_{G^{do(X=g(\bar{f}_A(\mathbf{n})))}}(X)}^{do(X=g(\bar{f}_A(\mathbf{n})))}(\mathbf{n}), n_X) \\
&= g(\bar{f}_A(\mathbf{n}), n_X).
\end{aligned}
\tag{13}
$$

Finally, if $Y \notin \mathrm{De}(X)$, then trivially $\bar{f}_Y^{do(X=g(A))}(\mathbf{n}) = \bar{f}_Y(\mathbf{n})$ and $\bar{f}_Y^{do(X=g(\bar{f}_A(\mathbf{n})))}(\mathbf{n}) = \bar{f}_Y(\mathbf{n})$.

This establishes the inductive step: if the results holds for the first $j \geq |\mathbf{N}|$ variables with respect to $\trianglelefteq$, then it also holds for the variable $j + 1$, since its parents are among the first $j$ variables. $\qquad\square$

**Lemma 26** (Superiority and Paths). *If $X \succeq_Y^{\mathrm{det},a} W$, then all paths $W \dashrightarrow Y$ must include $X$.*

*Proof.* If $W \notin \mathrm{An}(Y)$, there are no paths from $W$ to $Y$ and the conclusion is vacuously true. We assume from now on that $W \in \mathrm{An}(Y)$. Assume, for the sake of contradiction, that there is a path $\pi \colon W \dashrightarrow A \to Y$ in $G$ without $X$, where $A$ is a parent of $Y$. Consider the SCM with graph $G$ and structural assignments and noise distributions given by:

$$
\begin{cases}
f_Y(A, \mathrm{Pa}(Y) \setminus A, N_Y) = 2A + N_Y \cdot \mathbf{1}_{>0}(\sum_{Z \in \mathrm{Pa}(Y) \setminus A} Z) \\
f_{C \in \pi \setminus W}(\mathrm{Pa}(C), N_C) = \mathsf{pr}_\pi(C) + N_C \cdot \mathbf{1}_{>0}(\sum_{Z \in \mathrm{Pa}(C) \setminus \mathsf{pr}_\pi(C)} Z) \\
f_W(\mathrm{Pa}(W), N_W) = N_W \cdot \mathbf{1}_{>0}(\sum_{Z \in \mathrm{Pa}(W)} Z) \\
f_{V \notin \pi}(\mathrm{Pa}(V), N_V) = N_V \cdot \mathbf{1}_{>0}(\sum_{Z \in \mathrm{Pa}(V)} Z) \\
N_V \sim \mathrm{Ber}(\frac{1}{2})
\end{cases},
$$

where $\mathbf{1}_{>0} \colon \mathbb{R} \to \{0, 1\}$ is the unit step function, which maps values larger than 0 to 1, and all non-positive values to 0. Then, $\bar{f}_Y^{do(W=1)}(\mathbf{0}) = 2\bar{f}_A^{do(W=1)}(\mathbf{0}) = 2$, while, for every $X$, we have $\bar{f}_Y^{do(X=X)}(\mathbf{0}) = 0$. That is, for the setting $\mathbf{n} = \mathbf{0}$, there is no intervention on $X$ that is better than $do(W = 1)$, which contradicts the antecedent. $\qquad\square$

**Proposition 4** (Conditional vs Atomic superiority). *Let $X, W, Y$ be nodes in a DAG $G$. Then $X$ is average conditional-interventionally superior to $W$ relative to $Y$ in $G$ if and only if $X$ is atomic-interventionally superior to $W$ relative to $Y$ in $G$. That is:*

$$
X \succeq_Y^c W \iff X \succeq_Y^{\mathrm{det},a} W. \tag{5}
$$

*Proof.* ($\Rightarrow$): Assume $X \succeq_Y^c W$. Let $\mathfrak{C} = (\mathbf{V}, \mathbf{N}, \mathcal{F}, p_\mathbf{N})$ be an SCM with causal graph $G$ and $\mathbf{m} \in R_\mathbf{N}$. Let $g^* = \arg\max_g \mathbb{E}_\mathbf{n} \bar{f}_Y^{do(X=g(\mathbf{Z}_X))}(\mathbf{n})$. Then, $\forall h$, $\mathbb{E}_\mathbf{n} \bar{f}_Y^{do(X=g^*(\mathbf{Z}_X))}(\mathbf{n}) \geq \mathbb{E}_\mathbf{n} \bar{f}_Y^{do(W=h(\mathbf{Z}_W))}(\mathbf{n})$. This holds in particular for $p_\mathbf{N} = \delta(\mathbf{m})$. Denoting by $\mathcal{F}(\mathbf{A}, \mathbf{B})$ the set of functions with domain $\mathbf{A}$ and codomain $\mathbf{B}$, we can then write:

$$
\forall h \in \mathcal{F}(R_{\mathbf{Z}_W}, R_W), \bar{f}_Y^{do(X=g^*(\bar{f}_{\mathbf{Z}_X}(\mathbf{m})))}(\mathbf{m}) \geq \bar{f}_Y^{do(W=h(\bar{f}_{\mathbf{Z}_W}(\mathbf{m})))}(\mathbf{m}),
$$

where we also used Lemma 25. Now, since every $w \in R_W$ can be attained from $\bar{f}_{\mathbf{Z}_W}(\mathbf{m})$ by an appropriately chosen $h$, then choosing $X^* = g^*(\bar{f}_{\mathbf{Z}_X}(\mathbf{m}))$ allows us to write:

$$
\forall w \in R_W, \bar{f}_Y^{do(X=X^*)}(\mathbf{m}) \geq \bar{f}_Y^{do(W=w)}(\mathbf{m}).
$$

This proves that $X \succeq_Y^{\mathrm{det},a} W$.

$\qquad\qquad\square_\Rightarrow$

($\Leftarrow$): Assume now that $X \succeq_Y^{\mathrm{det},a} W$. Let $p_\mathbf{N} \in \mathcal{P}(\mathbf{N})$ and $\mathcal{F}(G) = \{f_V \colon V \in G\}$. We want to show that $\max_g \mathbb{E}_\mathbf{n} \bar{f}_Y^{do(X=g(\mathbf{Z}_X))}(\mathbf{n}) \geq \max_h \mathbb{E}_\mathbf{n} \bar{f}_Y^{do(W=h(\mathbf{Z}_W))}(\mathbf{n})$. From Lemma 25, we can write this as $\max_g \mathbb{E}_\mathbf{n} \bar{f}_Y^{do(X=g(\bar{f}_{\mathbf{Z}_X}(\mathbf{n})))}(\mathbf{n}) \geq \max_h \mathbb{E}_\mathbf{n} \bar{f}_Y^{do(W=h(\bar{f}_{\mathbf{Z}_W}(\mathbf{n})))}(\mathbf{n})$. Denote the expected value in the left-hand-side by $\alpha(g)$, and the one on the right-hand-side by $\beta(h)$. Assume, for the sake of contradiction, that there is $h^*$ such that $\beta(h^*) > \alpha(g)$ for all $g$. Define $H(\mathbf{n}) = h^*(\bar{f}_{\mathbf{Z}_W}(\mathbf{n}))$. Now, if $W \notin \mathrm{An}(Y)$, we simply define $g^*$ to output the observational value of $X$. If instead $W \in \mathrm{An}(Y)$, from Lemma 26, we know that[13] $X \in \mathrm{De}(W)$ and all paths from $W$ to $Y$ go through $X$. We then define $g^*(\bar{f}_{\mathbf{Z}_X}(\mathbf{n})) = \bar{f}_X[W](h^*(\bar{f}_{\mathbf{Z}_W}(\mathbf{n})), \mathbf{n})$. Let $G(\mathbf{n}) = g^*(\bar{f}_{\mathbf{Z}_X}(\mathbf{n}))$. Then:

$$
\begin{aligned}
\alpha(g^*) &= \mathbb{E}_\mathbf{n} \bar{f}_Y^{do(X=G(\mathbf{n}))}(\mathbf{n}) \\
&= \mathbb{E}_\mathbf{n} \bar{f}_Y[X](G(\mathbf{n}), \mathbf{n}) \\
&= \mathbb{E}_\mathbf{n} \bar{f}_Y[X](\bar{f}_X[W](H(\mathbf{n}), \mathbf{n}), \mathbf{n}) \\
&= \mathbb{E}_\mathbf{n} \bar{f}_Y[W](H(\mathbf{n}), \mathbf{n}) \\
&= \mathbb{E}_\mathbf{n} \bar{f}_Y^{do(W=H(\mathbf{n}))}[W](\mathbf{n}) \\
&= \beta(h^*).
\end{aligned}
$$

where in the fourth equality we used Lemma 37. This contradicts our assumption.

$\square_{\Leftarrow}$

$\square$

As mentioned in the main text (Section 3), the superiority relation for atomic interventions in non-deterministic (general) SCMs defined in the natural way is *not* equivalent to $\succeq_Y^c$. Indeed, consider the following example:

*Example* 27. Consider the SCM given by $Y = A \oplus W$, $A = Z \oplus W$ and $N_Z, N_W \sim \text{Bern}(1/2)$, where $\oplus$ is the XOR operator and all variables are binary. Setting $Z$ to 1 ensures that $Y = 1$, so that $\mathbb{E}_\mathbf{n} \bar{f}_Y^{do(Z=1)} = 1$. No atomic intervention on $A$ would accomplish this: $\mathbb{E}_\mathbf{n} \bar{f}_Y^{do(A=0)} = \mathbb{E}_\mathbf{n} \bar{f}_Y^{do(A=1)} = \frac{1}{2}$. Hence $A \not\succeq_Y^a Z$. However, $\mathbb{E}_\mathbf{n} \bar{f}_Y^{do(A=g(W))} = 1 = \max R_Y$ if one uses the policy $g(0) = 1$, $g(1) = 0$. Thus $A \succeq_Y^c Z$.

# D. Intervention Superiority Relations are Preorders

**Proposition 28.** *The interventional superiority relation between nodes is a preorder in $G$. The interventional superiority relation between node sets is also a preorder.*

*Proof.* Let $G$ be a DAG and let $Y \in G$. We will first prove the result for the interventional superiority relation on nodes.
Reflexivity: Let $X$ be a node in $G$ and $\mathfrak{C} \in \mathfrak{C}(G)$. For each setting $n$, the largest value of $Y$ that can be achieved by intervening on $X$ is attained when setting $X$ to $X^*(n) = \arg\max_X \bar{f}_Y^{do(X=X)}(n)$. Hence, $\bar{f}_Y^{do(X=X^*(n))}(n) \geq \bar{f}_Y^{do(X=X)}(n)$ for all $X \in R_X$, so that $X \succeq_Y^{\det,a} X$.
Transitivity: assume that $Z \succeq_Y^{\det,a} W$ and $W \succeq_Y^{\det,a} X$. Let $\mathfrak{C} \in \mathfrak{C}(G)$ and $n \in R_N$. Then $\max_X \bar{f}_Y^{do(X=X)}(n) \leq \max_w \bar{f}_Y^{do(W=w)}(n) \leq \max_Z \bar{f}_Y^{do(Z=Z)}(n)$. Hence $Z \succeq_Y^{\det,a} X$.
This establishes that $\succeq_Y^{\det,a}$ is a preorder in $G$. We now show the result for node sets. Let $\mathbf{X}$, $\mathbf{W}$ and $\mathbf{Z}$ be sets of nodes in $G$.
Reflexivity: let $X \in \mathbf{X}$. Since, by reflexivity of $\succeq_Y^{\det,a}$ on nodes, we have that $X \succeq_Y^{\det,a} X$, it trivially follows that $\mathbf{X} \succeq_Y^{\det,a} \mathbf{X}$.
Transitivity: assume that $\mathbf{Z} \succeq_Y^{\det,a} \mathbf{W}$ and $\mathbf{W} \succeq_Y^{\det,a} \mathbf{X}$. Let $X \in \mathbf{X}$. Then there is $W \in \mathbf{W}$ such that $W \succeq_Y^{\det,a} X$. There is also $Z \in \mathbf{Z}$ such that $Z \succeq_Y^{\det,a} W$. By transitivity of $\succeq_Y^{\det,a}$ on nodes, it follows that $Z \succeq_Y^{\det,a} X$. Hence $\mathbf{Z} \succeq_Y^{\det,a} \mathbf{X}$. $\square$

*Remark* 29 (Interventional Superiority is not an order, and it is not total). One may have expected interventional superiority (both on nodes and on node sets) to be a partial order in $G$. However, they are merely preorders. That is, the antisymmetry property does not hold. To see this for $\succeq_Y^{\det,a}$ on nodes, just notice that, if $X, W \notin \text{An}(Y)$, then trivially $X \succeq_Y^{\det,a} W$ and $W \succeq_Y^{\det,a} X$, no matter what $X$ and $W$ are. For node sets, consider the case where $X \subsetneq W$, but the best intervention lies in $X$. Then $X \succeq_Y^{\det,a} W$ and $W \succeq_Y^{\det,a} X$, even though $\mathbf{X} \neq \mathbf{W}$.
Notice also that $\succeq_Y^{\det,a}$ on nodes cannot be a total preorder: just consider the graph $A_1 \to Y \leftarrow A_2$. Once can have an SCM $\mathfrak{C}$ in which intervening on $A_1$ can lead to larger values of $Y$ than interventions on $A_2$. But one can also switch the structural assignments assignments of $\mathfrak{C}$, which would lead to the opposite conclusion. This example also shows that $\succeq_Y^{\det,a}$ on node sets also cannot be a total preorder.

# E. Proofs for The Minimal Globally Interventionally Superior Set

## E.1. Uniqueness of the mGISS

**Lemma 30** (Elements of a mGISS are not Comparable). *Let $\mathbf{A} \subseteq \mathbf{V}$ be a mGISS relative to $Y$. Let $X, X' \in \mathbf{A}$ and $X \neq X'$. Then $X' \not\succeq_Y^{\det,a} X$.*

*Proof.* Assume $X' \succeq_Y^{\det,a} X$ for the sake of contradiction. We will show that this implies that $\mathbf{A} \setminus X$ is also a GISS. That is, that for every element of $(\mathbf{V} \setminus Y) \setminus (\mathbf{A} \setminus X)$ there is an element of $\mathbf{A} \setminus X$ which is superior to it. Let $W \in (\mathbf{V} \setminus Y) \setminus (\mathbf{A} \setminus X)$. If $W = X$, then $X' \in \mathbf{A} \setminus X$ and $X' \succeq_Y^{\det,a} X$. If $W \neq X$, then $W \in (\mathbf{V} \setminus Y) \setminus \mathbf{A}$. Since $\mathbf{A}$ is a GISS, we can pick $\tilde{X} \in \mathbf{A}$ such that $\tilde{X} \succeq_Y^{\det,a} W$. In case $\tilde{X} = X$, we can choose instead $X'$. Indeed, since $X' \succeq_Y^{\det,a} X$ and $X \succeq_Y^{\det,a} W$, we have by transitivity of $\succeq_Y^{\det,a}$ (Proposition 28) that $X' \succeq_Y^{\det,a} W$. This shows that $\mathbf{A} \setminus X \subseteq \mathbf{A}$ is a GISS, contradicting the minimality of $\mathbf{A}$. $\square$

**Proposition 8** (Uniqueness of the mGISS). *Let $G$ be a DAG and $Y$ a node of $G$. The minimal globally interventionally superior set of $G$ relative to $Y$ is unique. We denote it by* $\mathrm{mGISS}_Y(G)$

*Proof.* Let $\mathbf{A}$ and $\mathbf{B}$ be minimal globally interventionally superiot sets of $G$ with respect to $Y$. Assume, for the sake of contradiction, that $\mathbf{B} \neq \mathbf{A}$. By minimality of $\mathbf{A}$, we have $\mathbf{B} \not\subseteq \mathbf{A}$, so that $\mathbf{B} \setminus \mathbf{A} \neq \emptyset$. Let $X \in \mathbf{B} \setminus \mathbf{A}$. In particular, $X \in (\mathbf{V} \setminus Y) \setminus \mathbf{A}$. Hence, $\exists Z \in \mathbf{A}$ s.t. $Z \succeq_Y^{\mathrm{det},a} X$. Either $Z \in \mathbf{A} \cap \mathbf{B}$ or $Z \in \mathbf{A} \setminus \mathbf{B}$. If $Z \in \mathbf{A} \setminus \mathbf{B}$, then in particular $Z \in (\mathbf{V} \setminus Y) \setminus \mathbf{B}$. Since $\mathbf{B}$ is a GISS, there is $X' \in \mathbf{B}$ such that $X' \succeq_Y^{\mathrm{det},a} Z$. By transitivity of $\succeq_Y^{\mathrm{det},a}$ (Proposition 28), it follows that $X' \succeq_Y^{\mathrm{det},a} X$. Similarly, if $Z \in \mathbf{A} \cap \mathbf{B}$, one again has two elements $Z$ and $X$ of $B$ such that $Z \succeq_Y^{\mathrm{det},a} X$. In both cases, this contradicts the assumption that $B$ is a GISS, as per Lemma 30. $\qquad\square$

### E.2. The LSCA Closure and $\Lambda$-structures

It will be useful to know that, in order to show that a node belongs to $\mathcal{L}^\infty(\mathbf{U})$, it suffices to prove that it belongs to the LSCA closure of a subset of $\mathbf{U}$. We show by induction that this is indeed the case.

**Lemma 31.** *If* $\mathbf{U}' \subseteq \mathbf{U}$*, then* $\mathcal{L}^\infty(\mathbf{U}') \subseteq \mathcal{L}^\infty(\mathbf{U})$*.* $\mathbf{U}' \subseteq \mathbf{U}$

*Proof.* Recall that $\mathcal{L}^\infty(\mathbf{U}) = \mathcal{L}^i(\mathbf{U})$ for some $i \in \mathbb{N}$. We will show the result by induction on $i \in \mathbb{N}$. The base case holds trivially: $\mathcal{L}^0(\mathbf{U}') = \mathbf{U}' \subseteq \mathbf{U} = \mathcal{L}^0(\mathbf{U})$. Now assume that $\mathcal{L}^i(\mathbf{U}') \subseteq \mathcal{L}^i(\mathbf{U})$ for a given $i \in \mathbb{N}$ (induction hypothesis). Let $V \in \mathrm{LSCA}(\mathcal{L}^i(\mathbf{U}'))$. Then there are paths $V \dashrightarrow X$, $V \dashrightarrow Y$ with $X, Y \in \mathcal{L}^i(\mathbf{U}')$ not containing $Y$ and $X$, respectively. But $X, y$ are also in $\mathcal{L}^i(\mathbf{U})$, so that $V \in \mathrm{LSCA}(\mathcal{L}^i(\mathbf{U}))$. Then $\mathrm{LSCA}(\mathcal{L}^i(\mathbf{U})) \subseteq \mathrm{LSCA}(\mathcal{L}^i(\mathbf{U}))$. Using once more the induction hypothesis, it follows that $\mathcal{L}^{i+1}(\mathbf{U}') = \mathrm{LSCA}(\mathcal{L}^i(\mathbf{U}')) \cup \mathcal{L}^i(\mathbf{U}') \subseteq \mathrm{LSCA}(\mathcal{L}^i(\mathbf{U})) \cup \mathcal{L}^i(\mathbf{U}) = \mathcal{L}^{i+1}(\mathbf{U})$. $\quad\square$

**Lemma 32.** *Let* $\mathbf{U} \subseteq \mathbf{V}$*. If* $V \in \mathrm{LSCA}(\mathbf{U}) \setminus \mathbf{U}$*, then $V$ forms a $\Lambda$-structure over* $(\mathbf{U}, \mathbf{U})$*.*

*Proof.* Let $V \in \mathrm{LSCA}(\mathbf{U}) \setminus \mathbf{U}$. By Definition 10, there are distinct $U, U' \in \mathbf{U}$ for which there are paths $\pi: V \dashrightarrow U$ and $\pi': V \dashrightarrow U'$ whose interiors do not intersect $\{U, U'\}$. Now, let $W$ (respectively $W'$) be the first element in $\pi$ (respectively $\pi'$) in $\mathbf{U}$. Notice that $W \neq W'$, otherwise $W = W'$ would be in $\mathrm{SCA}(U, U')$ and be reachable from $V$, so that $V$ would not be a minimal element of $\mathrm{SCA}(U, U')$. This would contradict $V \in \mathrm{LSCA}(U, U')$. Similarly, the paths $\pi|_W: V \dashrightarrow W$, $\pi'|_{W'}: V \dashrightarrow W'$ resulting from restricting $\pi$ cannot have interior intersections: such an intersection node $\tilde{V}$ would be an SCA of $U, U'$ reachable from $V$, so that $V \notin \mathrm{LSCA}(U, U')$ — again a contradiction. Therefore, $V$ forms a $\Lambda$-structure over $W, W'$.

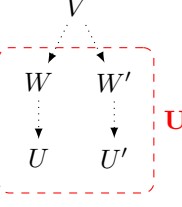

$\qquad\square$

**Theorem 15** (Simple Graphical Characterization of LSCA Closure). *A node $V \in \mathbf{V}$ is in the LSCA closure $\mathcal{L}^\infty(\mathbf{U})$ of $\mathbf{U} \subseteq \mathbf{V}$ if and only if $V$ forms a $\Lambda$-structure over $(\mathbf{U}, \mathbf{U})$. I.e.* $\mathcal{L}^\infty(\mathbf{U}) = \Lambda(\mathbf{U}, \mathbf{U})$*.*

*Proof.* Proof of $\subseteq$: If $\mathcal{L}^\infty(\mathbf{U}) = \mathbf{U}$, then the result is trivially true. We assume from now on that $\mathcal{L}^\infty(\mathbf{U}) \supsetneq \mathbf{U}$. We will prove that $V \in \mathcal{L}^\infty(\mathbf{U}) \Rightarrow V \in \Lambda(\mathbf{U}, \mathbf{U})$ by induction with respect to a chosen strict reverse topological order $<$ (*i.e.* $V' \in \mathrm{An}(V) \setminus \{V\} \Rightarrow V < V'$). The base case is $V_0 \in \mathbf{U}$, since an element of $\mathbf{U}$ will be the first element of $\mathcal{L}^\infty(\mathbf{U})$ for any chosen $<$. In this case, we can simply take the trivial paths $\pi = \pi' = (V_0)$. Then $V_0 \in \Lambda(\mathbf{U}, \mathbf{U})$. Now, assume that $V \in \mathcal{L}^\infty(\mathbf{U}) \setminus \mathbf{U}$ and that the implication holds for all $W \in \mathcal{L}^\infty(\mathbf{U})$ such that $W < V$ (induction hypothesis). Let $W, W'$ be[14] distinct elements of $\mathcal{L}^\infty(\mathbf{U})$ such that $V \in \mathrm{LSCA}(W, W')$. In particular, $W, W' < V$. By Lemma 32 applied to $\{W, W'\}$, there are paths $V \xrightarrow{\alpha} W$, $V \xrightarrow{\alpha'} W'$ intersecting only at $V$. Furthermore, by the induction hypothesis we have that $W, w' \in \Lambda(\mathbf{U}, \mathbf{U})$, so that there are paths $W \xrightarrow{\pi_1} U_1$, $W \xrightarrow{\pi_2} U_2$, $W' \xrightarrow{\pi'_1} U'_1$, $W' \xrightarrow{\pi'_2} U'_2$ such that $U_1, U_2, U'_1, U'_2 \in \mathbf{U}$, $\pi_1 \cap \pi_2 = \{W\}$ and $\pi'_1 \cap \pi'_2 = \{w'\}$.

---

[14] Such $W, W'$ must exist by the definition of $\mathcal{L}^\infty(\mathbf{U})$ whenever $\mathcal{L}^\infty(\mathbf{U}) \supsetneq \mathbf{U}$.

Let $\mathbf{S} = (\alpha \cup \pi_1 \cup \pi_2) \cap (\alpha' \cup \pi_1' \cup \pi_2')$ and $\unlhd$ be a chosen topological order. If $\mathbf{S} = \emptyset$, we can just take $\gamma = \pi_1 \circ \alpha \colon V \dashrightarrow U_1$ and $\gamma' = \pi_1' \circ \alpha' \colon V \dashrightarrow U_1'$ to form a $\Lambda$-structure for $V$ over $(\mathbf{U}, \mathbf{U})$. Assume from now on that $\mathbf{S} \neq \emptyset$. Let $S$ be the first element of $\mathbf{S}$ with respect to $\unlhd$. Since $\alpha \cap \alpha' = \emptyset$, there are three options: either (i) $S \in \pi_i \cap \alpha' \setminus \{W'\}$ for some $i$; (ii) $S \in \pi_i' \cap \alpha \setminus \{W\}$ for some $i$; or (iii) $S \in \pi_i \cap \pi_j'$ for some $i, j$. By symmetry, we can restrict ourselves to the cases (i) and (iii): the argument for (i) will also hold for (ii). In both cases (i) and (ii) we have $S \in \pi_i$ for some $i \in \{1, 2\}$. Without loss of generality, assume $s \in \pi_2$.

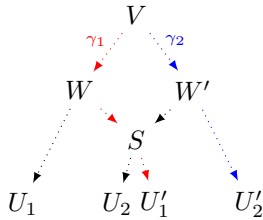

For case (iii), assume, also without loss of generality, that $s \in \pi_1'$. If furthermore $s \neq W'$, we can construct the following two paths with no non-trivial intersections:

$$\begin{cases} \gamma_1 = \pi_1'|^S \circ \pi_2|_s \circ \alpha : V \dashrightarrow U_1' \\ \gamma_2 = \pi_2' \circ \alpha' : V \dashrightarrow U_2' \end{cases} \tag{14}$$

To see that these paths have non non-trivial intersections, start by noticing that, by definition of $S$, there is no intersection between $\pi_2$ and $\pi_2'$ at nodes $A \lhd S$, so that $\pi_2|_S \cap \pi_2' = \emptyset$. And since $\pi_1' \cap \pi_2' = \{W'\}$ and $S \neq W'$, we have $\pi_1'|^S \cap \pi_2' = \emptyset$. Finally, $\pi_2 \cap \alpha' = \pi_2' \cap \alpha = \pi_1 \cap \alpha' = \emptyset$, since otherwise there would be elements of $S$ which are ancestors of $s$. Notice that this argument still holds if $S = W$, in which case $\gamma_1$ reduces to $\pi_1'|^W \circ \alpha$. This shows that $V \in \Lambda(\mathbf{U}, \mathbf{U})$ for case (iii), in case $S \neq W'$. If instead $S = W'$, we can simply choose paths similar to those for the case $S = W$ (just changing the numbers and the prime) as follows: $\gamma_1 = \pi_1 \circ \alpha$ and $\gamma_2 = \pi_2|^{W'} \circ \alpha'$.

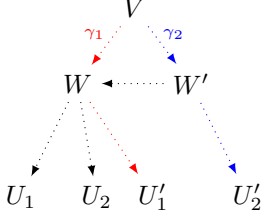
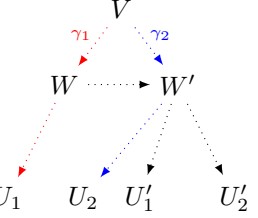

We now turn to case (i), where $S \in \pi_2 \cap \alpha' \setminus \{W\}$. Construct the paths:

$$\begin{cases} \gamma_1 = \pi_1 \circ \alpha : V \dashrightarrow U_1 \\ \gamma_2 = \pi_2|^s \circ \alpha'|_s : V \dashrightarrow U_2 \end{cases} \tag{15}$$

Notice that $\alpha \cap \pi_2|^S = \emptyset$, otherwise there would be a cycle in the DAG. Also, $\pi_1 \cap \alpha'|_S = \emptyset$ by definition of $S$. And trivially $\pi_1 \cap \pi_2|^S = \emptyset$ and $\alpha \cap \alpha' = \{V\}$.

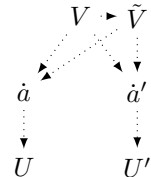

It follows that $\gamma_1$ and $\gamma_2$ intersect only trivially, so that $(v, \gamma_1, \gamma_2)$ forms a $\Lambda$-structure over $(\mathbf{U}, \mathbf{U})$.

$$\square_{\subseteq}$$

Proof of $\supseteq$: Let $V \in \Lambda(\mathbf{U}, \mathbf{U})$. Then, there is a pair of nodes $U, U' \in \mathbf{U}$ over which $V$ forms $\Lambda$-structures. We are going to show that $V \in \mathcal{L}^\infty(\{U, U'\})$. Let $\mathbf{L}$ be the set of all the $\Lambda$-structures $\lambda_i = (V, \pi_i \colon V \dashrightarrow U, \pi_i' \colon V \dashrightarrow U')$ over $(U, U')$. Let $A$ be the set of nodes in $\mathcal{L}^\infty(\{U, U'\})$ which belong to some $\pi_i$. Formally, $A = \{a \in \mathcal{L}^\infty(\{U, U'\}) \colon \exists i \text{ s.t. } \lambda_i \in \mathbf{L}, a \in \pi_i\} \setminus \{V\}$. Let $\dot{a}$ be the first element of $A$ with respect to a chosen topological order $\unlhd$. Denote by $\Pi'(\dot{a})$ the set of paths $\pi_i'$ belonging to some $\Lambda$-structure $(V, \pi_i', \pi_i)$ in $\mathbf{L}$ such that $\pi_i$ contains $\dot{a}$. Let $A'(\dot{a}) = \{a' \in \mathcal{L}^\infty(\{U, U'\}) \colon \exists \pi_i' \in \Pi'(\dot{a}) \text{ s.t. } a' \in \pi_i'\}$. Furthermore, let[15] $\dot{a}'$ be the first element of $A'(\dot{a})$ with respect to $\unlhd$. Denote by $(V, \dot{\pi}, \dot{\pi}')$ a $\Lambda$-structure of $\mathbf{L}$ such that $a \in \dot{\pi}$ and $a' \in \dot{\pi}'$. Notice that $\dot{a} \neq \dot{a}'$ and $\dot{\pi}|_{\dot{a}} \cap \dot{\pi}'|_{\dot{a}'} = \{V\}$, by definition of $\Lambda$-structure. In particular, $v \in \mathrm{SCA}(\dot{a}, \dot{a}')$. Suppose, for the sake of contradiction, that there is $\tilde{V} \in \mathrm{SCA}(\dot{a}, \dot{a}')$ such that $\tilde{V}$ is reachable from $V$. Then there is $\lambda = (V, \gamma, \gamma')$ in $\mathbf{L}$ such that $\tilde{V}, \dot{a} \in \gamma$. But $\tilde{V} \unlhd \dot{a}$, contradicting minimality of $\dot{a}$. Hence $V \in \mathrm{LSCA}(\dot{a}, \dot{a}')$. Finally, since $\dot{a}, \dot{a}' \in \mathcal{L}^\infty(\mathbf{U})$, it follows that $V \in \mathcal{L}^\infty(\mathbf{U})$.

---

[15]Notice that $A'$ (and $A$) are not empty (at least one of $\{U, U'\}$ is in $A'$ (and $A$).

$\square_{\supseteq}$

$\square$

### E.3. The LSCA Closure is the mGISS

**Lemma 33.** *Let $B \in \mathbf{V}$. Assume there are nodes $Z, W$ which are reachable from $B$ with paths whose interiors do not intersect $\mathcal{L}^{\infty}(\{Z, W\})$. Then $B \in \mathcal{L}^{\infty}(\{Z, W\})$.*

*Proof.* Notice that, since $Z, W \in \mathcal{L}^{\infty}(\{Z, W\})$, then $B \in \mathrm{SCA}(Z, W)$. If there are no SCAs of $(Z, W)$ reachable from $B$, then $B \in \mathrm{LSCA}(Z, W)$. Assume from now on that there are SCAs of $(Z, W)$ reachable from $B$. Let $S = \{B\} \cup \{\tilde{B} \in \mathrm{SCA}(Z, W) \colon \tilde{B} \text{ is reachable from } B\}$. Order $S$ with a chosen reverse topological order $\leq$, and denote its elements by $\tilde{B}_i$, where $i \leq j$ iff $\tilde{B}_i \leq \tilde{B}_j$. Remove from $S$ the elements $\tilde{B}_j$ for which every pair of paths from $\tilde{B}_j$ to $Z$ and $W$ intersects at some $\tilde{B}_k, k < j$. Denote the remaining nodes by $B_i, i \in 1, \ldots, M+1$. Let $\{L_k\}_k, k \in \{1, \ldots, K\}$ be a layering of $\{B_i\}_i$. Note that $B_1 \in L_1$. We will prove by induction on the layers that all the $B_i$ are in $\mathcal{L}^{\infty}(\{Z, W\})$. First, notice that all $A \in L_1$ must be in $\mathrm{LSCA}(Z, W)$; otherwise, there would be an SCA of $Z, W$ reachable from $A$ element of $S$ reachable by $A$ (and thus also from $B$) and (from Lemma 32) with non-intersecting paths to $Z$ and $W$ — hence, an element of $\{B_i\}$ reachable from $A$. This contradicts that $A \in L_1$. Hence, $L_1 \subseteq \mathcal{L}^{\infty}(\{Z, W\})$. This establishes the base case. Let $k \in \{2, \ldots, K\}$. Assume that $L_{k-1} \subseteq \mathcal{L}^{\infty}(\{Z, W\})$ (induction hypothesis). Let $A \in L_k$. Let $C, D$ be the first nodes in $\{B_i\}$ to which there are paths from $A$ not intersecting in $\{B_i\}$ (which must exist by construction of $\{B_i\}$). Clearly $A \in \mathrm{SCA}(C, D)$. But also $A \in \mathrm{LSCA}(C, D)$: if there was $\tilde{A} \in \mathrm{LSCA}(C, D)$ reachable from $A$, then in particular $\tilde{A} \in \{B_i\}$, contradicting the minimality of $C$ and $D$ with respect to $\leq$. Since, by the induction hypothesis, $C, D \in \mathcal{L}^{\infty}(\{Z, W\})$, then also $A \in \mathcal{L}^{\infty}(\{Z, W\})$. Finally, since all $B_i \in \mathcal{L}^{\infty}(\{Z, W\})$, by assumption $B$ has paths to $Z$ and $W$ never intersecting $\{B_i\}$, so that in particular $B \in \{B_i\}$ (and in fact $B = B_{M+1}$ and $B \in L_K$). Hence $B \in \mathcal{L}^{\infty}(\{Z, W\})$.

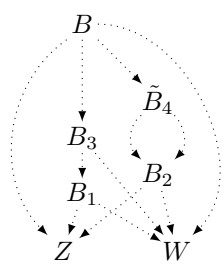

$\square$

**Lemma 34.** *Let $B \in \mathrm{An}(Y)$ and $B \notin \mathcal{L}^{\infty}(\mathrm{Pa}(Y))$. Then there is exactly one node $Z \in \mathcal{L}^{\infty}(\mathrm{Pa}(Y))$ reachable from $B$ by paths whose interiors do not contain elements from $\mathcal{L}^{\infty}(\mathrm{Pa}(Y))$.*

*Proof.* There must be at least one node in $\mathcal{L}^{\infty}(\mathrm{Pa}(Y))$ reachable from $B$ by paths not containing interior elements from $\mathcal{L}^{\infty}(\mathrm{Pa}(Y))$: since $\mathrm{Pa}(Y) \subseteq \mathcal{L}^{\infty}(\mathrm{Pa}(Y))$ and $B \in \mathrm{An}(Y)$, there are paths from $B$ to $Y$ crossing $\mathcal{L}^{\infty}(\mathrm{Pa}(Y))$ (in fact, paths must at least intersect $\mathrm{Pa}(Y)$). Choose one such path $\pi \colon B \dashrightarrow Y$. Let $Z$ be the first element of $\mathcal{L}^{\infty}(\mathrm{Pa}(Y))$ in $\pi$. Then the path $\pi|_Z \colon B \dashrightarrow Z$ obtained from $\pi$ by truncating it at $Z$ has no interior nodes in the closure $\mathcal{L}^{\infty}(\mathrm{Pa}(Y))$. Furthermore, if there would be a second path from $B$ to $W \in \mathcal{L}^{\infty}(\mathrm{Pa}(Y)) \setminus \{Z\}$ containing no interior nodes from the closure, then, by Lemma 33, $B$ would be in $\mathcal{L}^{\infty}(\{Z, W\})$ and thus in $\mathcal{L}^{\infty}(\mathrm{Pa}(Y))$ — contradiction. This establishes uniqueness. $\square$

**Corollary 35.** *Under the assumptions of Lemma 34, all paths from $B$ to $Y$ must go through $Z$.*

*Proof.* If there was a path from $B$ to $Y$ not containing $Z$, it would have to go through a parent $A$ of $Y$. But the first element of $\mathcal{L}^{\infty}(\mathrm{Pa}(Y))$ in this path (perhaps $A$ itself) would contradict the uniqueness of $Z$ from Lemma 34. $\square$

The following lemma relates blocked unrolled assigments with atomic interventions, and will be used to prove Theorem 16.

**Lemma 36.** *Let $X \in \mathbf{V}$ and $Y \in \mathbf{V} \cup \mathbf{N}$. Then $\bar{f}_Y[X](x, \mathbf{n}) = \bar{f}_Y^{do(X=x)}(\mathbf{n})$.*

*Proof.* Let $X$ be an endogenous variable. We want to prove that the expression holds for any variable $Y$. We will prove this by induction. Let $\trianglelefteq$ be a topological order on the nodes of $G^*$. Note that the first elements with respect to this order are the exogenous variables, *i.e.* $N \trianglelefteq Z$ whenever $N \in \mathbf{N}$ and $Z \in \mathbf{V}$. The result is true for the exogenous variables. Indeed, for $Y \in \mathbf{N}$ we have that $\bar{f}_Y[X](x, \mathbf{n}) = \bar{f}_Y(\mathbf{n}) = Y = \bar{f}_Y^{do(X=x)}(\mathbf{n})$, since $Y \notin \mathrm{De}(X) \cup \{X\}$ and $Y$ is exogenous (both in the pre- and post-intervention structural causal models). This establishes the base case of the induction. Now

let $Y$ be endogenous. For the inductive step, we will prove that, if the result is true for the parents $\mathrm{Pa}_{G^*}(Y)$ of $Y$ in $G^*$ (induction hypothesis), then it is also true for $Y$. Assume the antecedent (induction hypothesis). There are three possibilities: $Y \in \mathrm{De}(X) \setminus \{X\}$, $Y = X$ or $Y \notin \mathrm{De}(X)$. In case $Y \in \mathrm{De}(X) \setminus \{X\}$:

$$
\begin{aligned}
\bar{f}_Y[X](x, \mathbf{n}) &= f_Y(\bar{f}_{\mathrm{Pa}(Y)}[X](x, \mathbf{n}), n_Y). \\
&\overset{\text{I.H.}}{=} f_Y(\bar{f}^{do(X=x)}_{\mathrm{Pa}(Y)}(\mathbf{n}), n_Y) \\
&= f_Y^{do(X=x)}(\bar{f}^{do(X=x)}_{\mathrm{Pa}(Y)}(\mathbf{n}), n_Y) \\
&\overset{\text{def}}{=} \bar{f}_Y^{do(X=x)}(\mathbf{n}),
\end{aligned}
\tag{16}
$$

where in the third equality we used that $f_Y^{do(X=x)} = f_Y$. If instead $Y = X$, then one simply has $\bar{f}_Y[X](x, \mathbf{n}) \overset{\text{def}}{=} x$. Furthermore, $\bar{f}_Y^{do(X=x)}(\mathbf{n}) = \bar{f}_Y^{do(X=x)}(\mathbf{n}) = f_X^{do(X=x)}(\mathbf{n}) = x$, where the second equality holds simply because $X$ has no non-exogenous parents in the post-intervention graph. Finally, if $Y \notin \mathrm{De}(X)$, then $\bar{f}_Y[X](x, \mathbf{n}) = \bar{f}_Y(\mathbf{n})$ by definition. And $\bar{f}_Y^{do(X=x)}(\mathbf{n}) = f_Y^{do(X=x)}(\bar{f}^{do(X=x)}_{\mathrm{Pa}(Y)}(\mathbf{n}), n_Y) = f_Y(\bar{f}_{\mathrm{Pa}(Y)}(\mathbf{n}), n_Y)$, where in the last equality we used that $X \notin \mathrm{An}(Y) \Rightarrow \bar{f}^{do(X=x)}_{\mathrm{Pa}(Y)}(\mathbf{n}) = \bar{f}_{\mathrm{Pa}(Y)}(\mathbf{n})$. This establishes the inductive step: if the results holds for the first $j \geq |\mathbf{N}|$ variables with respect to $\trianglelefteq$, then it also holds for the variable $j + 1$, since its parents are among the first $j$ variables. $\qquad\square$

The following lemma shows how one can chain (blocked) unrolled assignments when there is a node $Z$ present in all paths from the blocking node $B$ to $Y$. This result is consistent with the intuition that, if all paths from $B$ to $Y$ must go through $Z$, then knowing the value of $Z$ is enough to compute $Y$.

**Lemma 37.** *If all paths from $B$ to $Y$ must include $Z$, then $\bar{f}_Y[B](b, \mathbf{n}) = \bar{f}_Y[Z](\bar{f}_Z[B](b, \mathbf{n}), \mathbf{n})$.*

*Proof.* Let $S$ be the poset whose elements are all the descendants $A$ of $B$ for which all paths from $B$ to $A$ must go through $Z$, and the partial order is a topological order $\trianglelefteq$. Denote the elements of $S$ by $W_i$, where $i \in \{0, \ldots, m-1\}$ corresponds to the position of $W_i$ in the order $\trianglelefteq$. We will prove the result by induction on a topological order. Notice that $Y \in S$. Thus, we can just show the result for all $W_i$. We start with the base case $W_0$. By definition:

$$
\bar{f}_{W_0}[Z](\bar{f}_Z[B](b, \mathbf{n}), \mathbf{n}) = f_{W_0}(\bar{f}_{\mathrm{Pa}(W_0)}[Z](\bar{f}_Z[B](b, \mathbf{n}), \mathbf{n}), n_{W_0}).
\tag{17}
$$

Recall that $\mathrm{Pa}(W_0) = (\mathrm{Pa}(W_0)_1, \ldots, \mathrm{Pa}(W_0)_{m_0})$. Hence, we want to check that $\bar{f}_{\mathrm{Pa}(W_0)_i}[Z](\bar{f}_Z[B](b, \mathbf{n}), \mathbf{n}) = \bar{f}_{\mathrm{Pa}(W_0)_i}[B](b, \mathbf{n})$ for all $i$, since in that case the right hand side of Equation (17) becomes $f_{W_o}(\bar{f}_{\mathrm{Pa}(W_0)_i}[B](b, \mathbf{n}), n_{W_0}) \overset{\text{def}}{=} \bar{f}_{W_0}[B](b, \mathbf{n})$.
If $\mathrm{Pa}(W_0)_i = Z$, then by definition of blocked unrolled assignment $\bar{f}_{\mathrm{Pa}(W_0)_i}[Z](\bar{f}_Z[B](b, \mathbf{n}), \mathbf{n}) = \bar{f}_Z[Z](\bar{f}_Z[B](b, \mathbf{n}), \mathbf{n}) = \bar{f}_Z[B](b, \mathbf{n})$.

If $\mathrm{Pa}(W_0)_i \neq Z$, then $\mathrm{Pa}(W_0)_i$ cannot be a descendant of $B$. Indeed, $W_0$ must have no parent that is a descendant of $B$, except maybe for $Z$. That is: $\mathrm{Pa}(W_0) \cap \mathrm{De}(B) \subseteq \{Z\}$. Otherwise, either that parent would be in $S$ and thus equal to $W_k$ for some $k > 0$, or it would be in $\mathrm{De}(B) \setminus (S \cup \{Z\})$, so that there would be a path $B \dashrightarrow W_0$ not crossing $Z$ — both cases contradict the definition of $W_0$. Hence, we only need to consider the case where $\mathrm{Pa}(W_0)_i \notin \mathrm{De}(B)$. In particular, $\mathrm{Pa}(W_0)_i \notin \mathrm{De}(Z)$. Then:

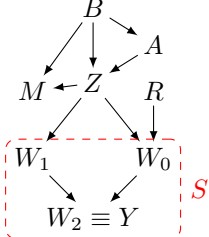

$$
\bar{f}_{\mathrm{Pa}(W_0)_i}[Z](\bar{f}_Z[B](b, \mathbf{n}), \mathbf{n}) = \bar{f}_{\mathrm{Pa}(W_0)_i}(\mathbf{n}) = \bar{f}_{\mathrm{Pa}(W_0)_i}[B](b, \mathbf{n}).
\tag{18}
$$

This shows the result for the base case $W_0$. Now, assume it to be true for all $W_j$ with $j \leq k$ (induction hypothesis). Equation (17) still holds for $W_{k+1}$. Now, each parent $\mathrm{Pa}(W_{k+1})_i$ must either be equal to $W_j$ for some $j < k + 1$, or not a descendant of $B$ (for the same reason as for the parents of $W_0$). In the latter case, Equation (18) still holds for $\mathrm{Pa}(W_{k+1})_i$. Hence, we only need to check that, for $\mathrm{Pa}(W_{k+1})_i = W_j$ (with $j < k + 1$), we have that $\bar{f}_{W_j}[Z](\bar{f}_Z[B](b, \mathbf{n})\mathbf{n}) = \bar{f}_{W_j}[B](b, \mathbf{n})$. But this is just the induction hypothesis. $\qquad\square$

**Theorem 16** (Superiority of the LSCA Closure). *Let $G$ be a causal graph and $Y$ a node of $G$ with at least one parent. Then, the LSCA closure $\mathcal{L}^\infty(\mathrm{Pa}(Y))$ of the parents of $Y$ is the minimal globally interventionally superior set $\mathrm{mGISS}(G)$ of $G$ relative to $Y$.*

*Proof.* We need to prove two results:

(i) $\mathcal{L}^\infty(\mathrm{Pa}(Y))$ is globally interventionally superior with respect to $Y$. That is: $\mathcal{L}^\infty(\mathrm{Pa}(Y)) \succeq_Y^{\det,a} \mathbf{V} \setminus (\mathcal{L}^\infty(\mathrm{Pa}(Y)) \cup \{Y\})$.

(ii) Furthermore, this is the minimal set with this property. Namely, removing any node $B$ from $\mathcal{L}^\infty(\mathrm{Pa}(Y))$ would result in a set $I = \mathcal{L}^\infty(\mathrm{Pa}(Y)) \setminus \{B\}$ that is not interventionally superior to $\mathbf{V} \setminus (I \cup \{Y\})$.

If $\mathrm{Pa}(Y) = \emptyset$, the theorem is vacuously true. Assume $\mathrm{Pa}(Y) \neq \emptyset$ from now on.

Proof of *(i)*: Let $B \in \mathbf{V} \setminus \mathcal{L}^\infty(\mathrm{Pa}(Y))$ and $B \neq Y$. We want to show that there is $A$ in the closure $\mathcal{L}^\infty(\mathrm{Pa}(Y))$ such that $A \succeq_Y^{\det,a} B$.

If $B$ is not an ancestor of $Y$, then trivially $\bar{f}_Y^{do(B=b)}(\mathbf{n}) = \bar{f}_Y(\mathbf{n})$ for all $\mathbf{n} \in R_{\mathbf{N}}$ and for all $b \in R_B$, so that in particular $\max_{b \in R_B} \bar{f}_Y^{do(B=b)}(\mathbf{n}) = \bar{f}_Y(\mathbf{n})$. Now, let $A$ be a parent of $Y$, and $a^* = \bar{f}_A(\mathbf{n})$ (*i.e.* $a^*$ is the value that $A$ would attain if no intervention was performed). Then, from the definition of unrolled assignment and atomic intervention, $\bar{f}_Y^{do(A=a^*)}(\mathbf{n}) = \bar{f}_Y(\mathbf{n})$. Thus, $\max_{a \in R_A} \bar{f}_Y^{do(A=a)}(\mathbf{n}) \geq \bar{f}_Y(\mathbf{n}) = \max_{b \in R_B} \bar{f}_Y^{do(B=b)}(\mathbf{n})$. That is, $A \succeq_Y^{\det,a} B$.

Assume from now on that $B$ is an ancestor of $Y$. From Lemma 34 there is one and only one node $Z \in \mathcal{L}^\infty(\mathrm{Pa}(Y))$ reachable from $B$ by paths not containing intermediate elements from $\mathcal{L}^\infty(\mathrm{Pa}(Y))$. Let $z^* \in \arg\max_{z \in R_Z}[\bar{f}_Y^{do(Z=z)}(\mathbf{n})]$. Further, Let $b \in R_B$. From Lemma 37, we have that $\bar{f}_Y[B](b, \mathbf{n}) = \bar{f}_Y[Z](\bar{f}_Y[B](b, \mathbf{n}), \mathbf{n})$, which of course is at most $\bar{f}_Y[Z](z^*, \mathbf{n})$. Finally, Lemma 36 allows us to relate this to a post-intervention unrolled assignment as $\bar{f}_Y[Z](z^*, \mathbf{n}) = \bar{f}_Y^{do(Z=z)}(\mathbf{n})$. This shows that $\max_{b \in R_B} \bar{f}_Y^{do(B=b)}(\mathbf{n}) \leq \max_{z \in R_Z} \bar{f}_Y^{do(Z=z)}(\mathbf{n})$, so that $Z \succeq_Y^{\det,a} B$.

$$\square_{(i)}$$

Proof of *(ii)*: We want to show that, for any causal graph $G$ and node $Y$ from $G$, removing any node[16] from $\mathcal{L}^\infty(\mathrm{Pa}(Y))$ will result in a set $I$ for which there is an SCM (with causal graph $G$) such that $I$ is not interventionally superior to $\bar{I} \setminus \{Y\}$, *i.e.* $I \not\succeq_Y^{\det,a} \bar{I} \setminus \{Y\}$. In other words, we want to prove that:

$$\forall \text{ DAG } G = (\mathbf{V}, E), \forall Y \in \mathbf{V}, \forall B \in \mathcal{L}^\infty(\mathrm{Pa}(Y)),$$
$$\exists \text{ SCM } \mathfrak{C} \text{ s.t. } G^{\mathfrak{C}} = G \text{ and } I = \mathcal{L}^\infty(\mathrm{Pa}(Y)) \setminus \{B\} \not\succeq_Y^{\det,a} \bar{I} \setminus \{Y\}. \tag{19}$$

Let $G$ be a DAG, $Y$ be a node of $G$ and $B$ an element of the closure $\mathcal{L}^\infty(\mathrm{Pa}(Y))$. Let also $I = \mathcal{L}^\infty(\mathrm{Pa}(Y)) \setminus \{B\}$. In particular, $B \in \bar{I} \setminus \{Y\}$. We will show that there is no element of $I$ which is interventionally superior to $B$, thus proving that $I \not\succeq_Y^{\det,a} \bar{I} \setminus \{Y\}$. We will divide the proof in two cases: $B \in \mathrm{Pa}(Y)$ and $B \in \mathcal{L}^\infty(\mathrm{Pa}(Y)) \setminus \mathrm{Pa}(Y)$.

Assume $B \in \mathrm{Pa}(Y)$. We can construct an SCM with causal graph $G$ as follows:

$$\begin{cases} f_Y(\mathrm{Pa}(Y), N_Y) = 2B + \mathbf{1}_{>0}\left(\sum_{W \in \mathrm{Pa}(Y) \setminus \{B\}} W\right) + N_Y \\ f_B(\mathrm{Pa}(B), N_B) = N_B \cdot \left(1 - \mathbf{1}_{>0}\left(\sum_{W \in \mathrm{Pa}(B)} W\right)\right) \\ f_{V \neq Y, B}(\mathrm{Pa}(V), N_V) = \mathbf{1}_{>0}\left(\sum_{W \in \mathrm{Pa}(V)} W\right) + N_V \\ N_{V \neq B} \sim \delta(0) \\ N_B \sim \mathrm{Ber}(1/2) \end{cases}, \tag{20}$$

where all endogenous variables are binary except for $Y$ (whose range is $\mathbb{N}$), and all exogenous variables are simply zero except for $N_B$, which is also binary. The idea is that $B$ has a stronger influence on $Y$ than all the other parents of $Y$ combined, and there are values of $\mathbf{n}$ (namely whenever $N_B = 0$) for which $B$ is not influenced by other variables. We need to show that, for all $X \in I$, there is $\mathbf{n} \in R_{\mathbf{N}}$ such that

$$\max_{x \in R_X} \bar{f}_Y^{do(X=x)}(\mathbf{n}) < \max_{b \in R_B} \bar{f}_Y^{do(B=b)}(\mathbf{n}). \tag{21}$$

---

[16]It is enough to remove a single node: if removing one node results in a non-interventionally superior set, removing more nodes could clearly never result in an interventionally superior set.

Notice that $R_{\mathbf{N}} = \{\mathbf{0}, \mathbf{e}_{N_B}\}$, where $\mathbf{e}_{N_B}$ is zero everywhere except for the $N_B$ element, which is $1$. Let $X \in I$ and choose $\mathbf{n} = \mathbf{0}$. We have $\max_{b \in \{0,1\}} \bar{f}_Y^{do(B=b)}(\mathbf{0}) = \bar{f}_Y^{do(B=1)}(\mathbf{0}) = 2 + \mathbf{1}_{>0}\left(\sum_{W \in \mathrm{Pa}(B)} W\right) \geq 2$. Furthermore:

$$
\begin{aligned}
\max_{x \in \{0,1\}} \bar{f}_Y^{do(X=x)}(\mathbf{0}) &= \max_{x \in \{0,1\}} \left( 2n_B \cdot \left(1 - \mathbf{1}_{>0}\left(\sum_{W \in \mathrm{Pa}(B)} W\right)\right) + \mathbf{1}_{>0}\left(\sum_{W \in \mathrm{Pa}(Y) \setminus \{B\}} W\right)\right) \\
&= \max_{x \in \{0,1\}} \left( 0 + \mathbf{1}_{>0}\left(\sum_{W \in \mathrm{Pa}(Y) \setminus \{B\}} W\right)\right) \\
&\leq 1 < 2 \leq \max_{b \in \{0,1\}} \bar{f}_Y^{do(B=b)}(\mathbf{0}).
\end{aligned}
\tag{22}
$$

This proves the result for $B \in \mathrm{Pa}(Y)$.
Assume now that $B \in \mathcal{L}^\infty(\mathrm{Pa}(Y)) \setminus \mathrm{Pa}(Y)$. From Theorem 15, there are nodes $A_1, A_2 \in \mathrm{Pa}(Y)$ which are reachable from $B$ by paths $\pi_1, \pi_2$ which only intersect at $B$. Denote by $\mathrm{pr}_i$ the operator which, given a node $A$ in the path $\pi_i$ different from $B$, outputs the previous node in that path. We construct an SCM with causal graph $G$ as follows:

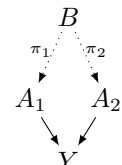

$$
\begin{cases}
f_Y(A_1, A_2, \mathrm{Pa}_Y \setminus \{A_1, A_2\}, N_Y) = 2A_1 \cdot A_2 + \mathbf{1}_{>0}\left(\sum_{W \in \mathrm{Pa}(Y) \setminus \{A_1, A_2\}} W\right) + N_Y \\
f_B(\mathrm{Pa}(B), N_B) = N_B \cdot \left(1 - \mathbf{1}_{>0}\left(\sum_{W \in \mathrm{Pa}(B)} W\right)\right) \\
f_{A \in \pi_i \setminus \{B\}}(\mathrm{pr}_i(A), \mathrm{Pa}(A) \setminus \{\mathrm{pr}_i(A)\}, N_A) = \mathrm{pr}_i(A) + N_A \mathbf{1}_{>0}\left(\sum_{W \in \mathrm{Pa}(A) \setminus \{\mathrm{pr}_i(A)\}} W\right) \\
f_{V \notin \pi_1 \cup \pi_2 \cup \{Y\}}(\mathrm{Pa}(V), N_V) = \mathbf{1}_{>0}\left(\sum_{W \in \mathrm{Pa}(V)} W\right) + N_V \\
N_{V \neq B} \sim \delta(0) \\
N_B, N_{A \in \pi_i} \sim \mathrm{Ber}(1/2)
\end{cases}
\tag{23}
$$

where again all endogenous variables except $Y$ are binary, and all exogenous variables are zero except for those of the type $N_A, A \in \pi_i$, which is also binary. Let $X \in I$. We again need to show that there is $\mathbf{n} \in R_{\mathbf{n}}$ such that Equation (21) holds. One again chooses the setting $\mathbf{n} = \mathbf{0}$. The intuition behind this SCM is similar to that of Equation (20), with the added property that the elements of the paths $\pi_i$ are simply noisy copies of $B$, and perfect copies when $\mathbf{N} = \mathbf{0}$. In particular, $A_i = B, i \in \{1, 2\}$, or, using the language of unrolled assignments, $\bar{f}_{A_i}(\mathbf{0}) = \bar{f}_B(\mathbf{0})$. Notice that $\bar{f}_{A_1}(\mathbf{0}) \cdot \bar{f}_{A_2}(\mathbf{0}) = \bar{f}_B(\mathbf{0})^2 = \bar{f}_B(\mathbf{0})$, since $B$ is binary. These equalities still hold in the SCMs resulting from atomically intervening on $B$. Hence:

$$
\bar{f}_Y^{do(B=b)}(\mathbf{0}) = 2b + \mathbf{1}_{>0}\left(\sum_{W \in \mathrm{Pa}(Y) \setminus \{A_1, A_2\}} \bar{f}_W(\mathbf{0})\right) \geq 2b.
\tag{24}
$$

Now, if $X = A \in \pi_1 \setminus \{B\}$, then $A_1$ is a perfect copy of $A$ while $A_2$ is still a perfect copy of $B$. Hence:

$$
\begin{aligned}
\bar{f}_Y^{do(A=a)}(\mathbf{0}) &= 2 \underbrace{\bar{f}_{A_1}^{do(A=a)}(\mathbf{0})}_{a} \cdot \underbrace{\bar{f}_{A_2}^{do(A=a)}(\mathbf{0})}_{0} + \mathbf{1}_{>0}\left(\sum_{W \in \mathrm{Pa}(Y) \setminus \{A_1, A_2\}} \bar{f}_W(\mathbf{0})\right) \\
&= \mathbf{1}_{>0}\left(\sum_{W \in \mathrm{Pa}(Y) \setminus \{A_1, A_2\}} \bar{f}_W(\mathbf{0})\right) \leq 1 < 2 \leq \bar{f}_Y^{do(B=1)}(\mathbf{0})
\end{aligned}
\tag{25}
$$

The same argument holds if $X = A \in \pi_2 \setminus \{B\}$.
Finally, if instead $X \notin \pi_1 \cup \pi_2 \cup \{Y\}$, then $\bar{f}_Y^{do(X=x)}(\mathbf{0}) = 0 + \mathbf{1}_{>0}\left(\sum_{X \in \mathrm{Pa}(Y) \setminus \{A_1, A_2\}} \bar{f}_X(\mathbf{0})\right) \leq 1 < 2 \leq \bar{f}_Y^{do(B=1)}(\mathbf{0})$, where the first equality holds because, for $\mathbf{n} = \mathbf{0}$, intervening on $X$ does not affect the elements of the $\pi_i$, including the $A_i$.

$\square_{(ii)}$

$\square$

## F. C4 Proofs

**Definition 38.** *We define the following additional notation and terminology.*

- *For any set of nodes $B$ and any node $v'$, a path $\pi_{v'}$ that ends in $v'$ is uninterrupted by $B$ iff $(\pi_{v'} \cap B) \backslash \{v'\} = \emptyset$.*

- *A $\Lambda$-structure which consists of a single node is called degenerate.*

- *For any set of nodes $B$, any $\Lambda$-structure over $(B, B)$ is referred to as a $\Lambda_B$-structure.*

- *$u \xleftarrow{\pi_u} v \xrightarrow{\pi_w} w$ denotes a $\Lambda$-structure $(v, \pi_u, \pi_w)$ with paths $\pi_u : v \dashrightarrow u$, $\pi_w : v \dashrightarrow w$. If the paths' names are not relevant or clear from the context, we write simply $u \dashleftarrow v \dashrightarrow w$.*

- *As an exception, in the proofs of this section we use lower case letters for nodes, as is customary in graph theory texts.*

**Lemma 39.** *Let $G = (V, E)$ be a DAG, $U \subseteq V$. $v \in \mathcal{L}^\infty(U)$ iff there exists a $\Lambda_{\mathcal{L}^\infty(U)}$-structure $v' \dashleftarrow v \dashrightarrow v^*$ for some $v', v^* \in \mathcal{L}^\infty(U)$.*

*Proof.* It is easily seen that $\mathcal{L}^\infty(\mathcal{L}^\infty(U)) = \mathcal{L}^\infty(U)$; therefore, this lemma is a direct corollary of Proposition 15. $\square$

**Lemma 40** (Existence of $\Lambda$-substructure). *Let $B$ be a set of nodes, and let $b_1, b_2 \in B$ s.t. $b_1 \neq b_2$. Let $v \notin \{b_1, b_2\}$ be a node and let $\pi_1 : v \dashrightarrow b_1$, $\pi_2 : v \dashrightarrow b_2$, s.t. $b_1 \notin \pi_2$ and $b_2 \notin \pi_1$ (note that we do not assume $\pi_1 \cap \pi_2 = \{v\}$, meaning that other overlaps remain possible). Then, in the subgraph consisting of the two paths (as in, the graph that includes all the nodes and all the edges that are in at least one of the paths), there exists a $\Lambda_B$-structure $b_1 \dashleftarrow v' \dashrightarrow b_2$ where $v' \in \pi_1 \cap \pi_2$.*

*Proof.* For $v' \in \arg \min_{\preccurlyeq} \pi_1 \cap \pi_2$, $b_1 \xleftarrow{\pi_1|^{v'}} v' \xrightarrow{\pi_2|^{v'}} b_2$ is a $\Lambda_B$-structure. $\square$

**Lemma 41.** *Let $G = (V, E)$ be a DAG, $U \subseteq V$, $v \in V$. If $v'$ is a $U$-connector of $v$, then $v'$ has a path from $v$ uninterrupted by $\mathcal{L}^\infty(U)$.*

*Proof.* On the one hand, $v' \in De(v)$ so it has some path from $v$. On the other hand, any path $\pi_{v'} = v \dashrightarrow v'$ is uninterrupted by $\mathcal{L}^\infty(U)$: otherwise there would be a node $v'' \neq v'$ s.t. $v'' \in \pi_{v'} \cap \mathcal{L}^\infty(U)$, but then $v' \preccurlyeq v''$ and $v'' \in De(v) \cap \mathcal{L}^\infty(U)$, violating $v' \in \arg \max_{\preccurlyeq} [De(v) \cap \mathcal{L}^\infty(U)]$. $\square$

**Lemma 18** (Uniqueness and Characterization of Connectors). *Let $G = (\mathbf{V}, E)$ be a DAG, $\mathbf{U} \subseteq \mathbf{V}$, $V \in \mathbf{V}$. If $V$ has a $\mathbf{U}$-connector $V'$, then $V'$ is the unique node for which there is a path $\pi_{V'} = V \dashrightarrow V'$ s.t. $\pi_{V'} \cap \mathcal{L}^\infty(\mathbf{U}) = \{V'\}$.*[17]

*Proof.* If $De(v) \cap \mathcal{L}^\infty(U) = \emptyset$ or if $v \in \mathcal{L}^\infty(U)$, the lemma is trivial. Assume $De(v) \cap \mathcal{L}^\infty(U) \neq \emptyset$ and $v \notin \mathcal{L}^\infty(U)$. Let $v' \in \arg \max_{\preccurlyeq} [De(v) \cap \mathcal{L}^\infty(U)]$; by Lemma 41, we know that there is a path $\pi_{v'} : v \dashrightarrow v'$ uninterrupted by $\mathcal{L}^\infty(U)$. We claim that there cannot exist another node $v^* \neq v'$ s.t. $v^* \in \mathcal{L}^\infty(U)$ has a path $\pi_{v^*} : v \dashrightarrow v^*$ uninterrupted by $\mathcal{L}^\infty(U)$. Assume for the sake of contradiction that such a node $v^*$ exists. Because both paths are uninterrupted by $\mathcal{L}^\infty(U)$, and $v', v^* \in \mathcal{L}^\infty(U)$, Lemma 40 implies the existence of a (non-degenerate since $v' \neq v^*$) $\Lambda_{\mathcal{L}^\infty(U)}$-structure $v^* \dashleftarrow \tilde{v} \dashrightarrow v'$ where $\tilde{v} \in \pi_{v'} \cap \pi_{v^*}$. Therefore, by Lemma 39, $\tilde{v} \in \mathcal{L}^\infty(U)$. However, $\tilde{v} \in \pi_{v'}$ and $\tilde{v} \neq v'$, so $\pi_{v'}$ is interrupted by $\mathcal{L}^\infty(U)$, which yields a contradiction. $\square$

**Theorem 19.** *C4 correctly computes $\mathcal{L}^\infty(U)$.*

*Proof.* We claim that the algorithm computes the connectors correctly: that is, we claim that for every $\mathfrak{v} \in V$, upon termination of the appropriate loop (meaning the loop where $v = \mathfrak{v}$ for $\mathfrak{v} \notin U$, or before the first loop begins for $\mathfrak{v} \in U$), $\mathfrak{c}[\mathfrak{v}] = \mathfrak{v}'$ iff $\mathfrak{v}'$ is the $U$-connector of $\mathfrak{v}$, and $\mathfrak{c}[\mathfrak{v}] = \text{NULL}$ iff $\mathfrak{v}$ has no $U$-connector. Note that our claim implies that for $\mathfrak{v} \in V$, upon termination of the appropriate loop, $\mathfrak{v} \in S \Leftrightarrow \mathfrak{v} \in \mathcal{L}^\infty(U)$: this is because $S$ is easily seen to include exactly the nodes for which $\mathfrak{c}[\mathfrak{v}] = \mathfrak{v}$, which by our claim are their own $U$-connectors, which holds iff $\mathfrak{v} \in \mathcal{L}^\infty(U)$. Note that once the appropriate loop terminates, $\mathfrak{c}[\mathfrak{v}]$ is never reassigned and $\mathfrak{v}$ is not added to or removed from $S$, so the connector of $\mathfrak{v}$ and its membership in $S$ or lack thereof remain correct through the end of the algorithm.

---

[17]If $V$ is its own connector, the path is trivial.

Let $v_1, \ldots, v_n$ be a reverse topological order of $V$ (no need to assume it is the one used in the algorithm's loop). Assume the claim is true for $v_1, \ldots, v_{i-1}$, and let us prove it is true for $v_i$. If $v_i \in U$, the claim is true by initialization of the algorithm; so assume $v_i \notin U$. There are three cases to consider in the loop's iteration:

1. $C = \emptyset$. In that case, the algorithm keeps $\mathfrak{c}[v_i] = \text{NULL}$. We claim that indeed $v_i$ has no connector, meaning that $De(v_i) \cap \mathcal{L}^\infty(U) = \emptyset$. First, no child of $v_i$ has a connector, meaning that no child of $v_i$ has a descendant in $\mathcal{L}^\infty(U)$. Thus, no node in $\mathcal{L}^\infty(U)$ is reachable from $v_i$. On the other hand, this also means that no reachable node from $v_i$ is in $U$, and we have also assumed that $v_i$ itself is not in $U$: thus, by Theorem 15, it follows that $v_i \notin \mathcal{L}^\infty(U)$. Since neither $v_i$ nor any node reachable from $v_i$ is in $\mathcal{L}^\infty(U)$, indeed $De(v_i) \cap \mathcal{L}^\infty(U) = \emptyset$.

2. $|C| = 1$. Let $x \in V$ s.t. $C = \{x\}$. The algorithm sets $\mathfrak{c}[v_i] = x$. We claim that indeed $x$ is the $U$-connector of $v_i$. By the inductive assumption and Lemma 18, $x$ is the unique element from $\mathcal{L}^\infty(U)$ reachable from $v_i$ via a *non-trivial* path uninterrupted by $\mathcal{L}^\infty(U)$, as any non-trivial path must go through a child, and we can apply the inductive assumption and Lemma 18 to each child. However, to show that $x$ is indeed the connector of $v_i$, we need to rule out the possibility of a trivial path to $\mathcal{L}^\infty(U)$, namely to rule out the possibility that $v_i \in \mathcal{L}^\infty(U)$. Since $v_i \notin U$, then by Proposition 15 it is sufficient to rule out the existence of a non-degenerate $\Lambda$-structure from $v_i$ to $U$. However, as we noted, any non-trivial path from $v_i$ to $U$ (and hence to $\mathcal{L}^\infty(U)$) must go through $x$, and hence any two paths to distinct nodes in $U$ must overlap at $x \neq v_i$, meaning that they do not make a $\Lambda$-structure.

3. $|C| > 1$. In that case, the algorithm sets $\mathfrak{c}[v_i] = v_i$. We claim that $v_i \in \mathcal{L}^\infty(U)$ (and so $v_i$ is its own connector). By Proposition 15, we need to establish the existence of a $\Lambda$-structure from $v_i$ to $U$. Since $|C| > 1$, then there exist $s_1, s_2 \in C$ s.t. $s_1 \neq s_2$, and there exist children $t_1, t_2$ of $V$ s.t. $\mathfrak{c}[t_1] = s_1$ and $\mathfrak{c}[t_2] = s_2$; by the inductive assumption, $s_1$ and $s_2$ are respectively the connectors of $t_1$ and $t_2$. Therefore, $s_1$ and $s_2$ are in $\mathcal{L}^\infty(U)$. By Lemma 18, there exist paths $\pi_1 = t_1 \dashrightarrow s_1$ and $\pi_2 = t_2 \dashrightarrow s_2$ uninterrupted by $\mathcal{L}^\infty(U)$. These paths do not overlap: had they overlapped, then by Lemma 40 they would've contained a $\Lambda$-substructure $s_1 \dashleftarrow z \dashrightarrow s_2$ s.t. $z \in \pi_1 \cap \pi_2$ so by Lemma 39 $z \in \mathcal{L}^\infty(U)$, making neither $\pi_1$ nor $\pi_2$ uninterrupted by $\mathcal{L}^\infty(U)$. Since $t_1$ and $t_2$ are children of $v_i$, we may prepend the edges $v_i \to t_1$ and $v_i \to t_2$ to $\pi_1$ and $\pi_2$ respectively and get paths $\pi_1' = v_i \to t_1 \dashrightarrow s_1$ and $\pi_2' = v_i \to t_2 \dashrightarrow s_2$; since $\pi_1$ and $\pi_2$ do not overlap, these two paths yield a $\Lambda$-structure from $v_i$ to $L^\infty(U)$, which by Lemma 39 implies $v_i \in L^\infty(U)$.

$\square$

**Theorem 20.** *C4 runs in $O(|\mathbf{V}| + |E|)$ time.*

*Proof.* If the graph is not given in adjacency list representation, we convert it to this representation in $O(|V| + |E|)$ time. Initialization in C4 is trivially $O(|V|)$. Reverse topological sorting can be done in $O(|V| + |E|)$ using Kahn's algorithm. In the loop, for each $v \in V \backslash U$, we go over all outgoing edges from $v$ to compute $C$, which because of the adjacency list representation takes $O(|\text{Ch}(v)|)$ time. In aggregate over the entire operation of the algorithm, computing $C$ takes $O(|E|)$ time overall, as each edge is inspected at most once. The loop runs $O(|V|)$ times, and all operations in it except the computation of $C$ take $O(1)$ time, so all steps except computing $C$ take at most $O(|V|)$ time overall. Thus, the running time of the algorithm is $O(|V| + |E|)$. $\square$

## G. Supplementary Material for Experimental Results

The results of the experiments testing our search space reduction method are presented in Figure 5 and Figure 6, for randomly generated graphs and real-world datasets, respectively.

All real-world datasets come from the `bnlearn` repository, except for the `railway` dataset, which was provided by ProRail, the institution responsible for traffic control in the Dutch railway system.

The `railway` dataset consists of a graph whose nodes represent train delays in a segment of the Dutch railway system, measured at specific "points of interest" (such as train stations). Each node is labeled with a code identifying the train, an acronym for the point of interest, a letter indicating the train's activity at that location—arriving (A), departing (V), or passing through (D)—and the planned time for that activity. Arrows are drawn between delay nodes that are known to influence each other. For example, arrows connect nodes of the same train at consecutive times, as the delay of a train at time $t$ will influence its delay at $t + \Delta t$. Additionally, arrows may connect nodes corresponding to train activities sharing

the same platform, since a train must wait for the preceding train to vacate the platform before using it. This dataset can be found in the code repository which supplements this paper.

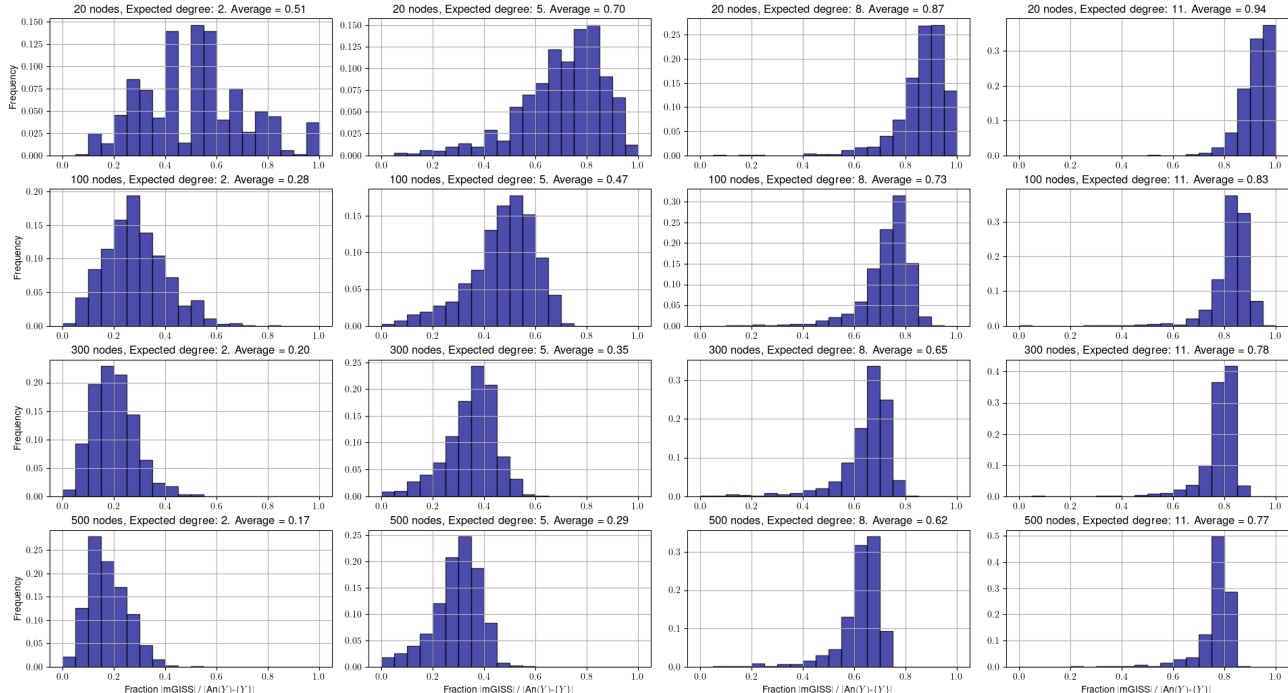

Figure 5: Fraction of nodes remaining after applying our search space filtering procedure, on random graphs. 1000 graphs were generated for each pair (number of nodes, expected degree). The impact of our method decreases with the expected degree, and increases with the number of nodes.

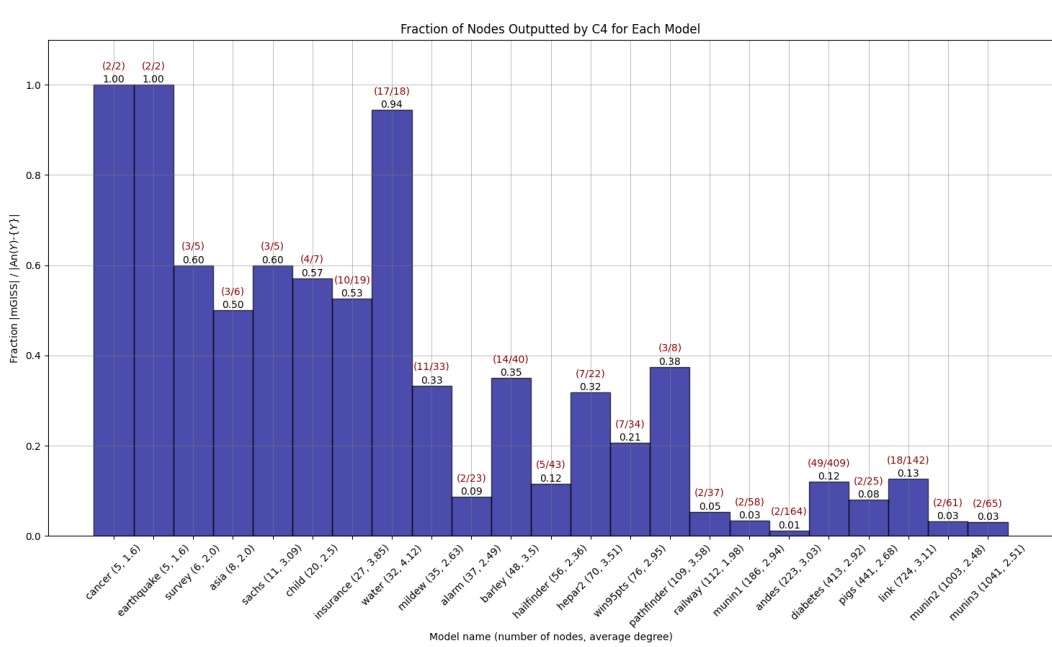

Figure 6: Fraction of nodes remaining after applying our search space filtering procedure, on real-world graphs. All models come from the `bnlearn` repository except for the `railway` model. The models are sorted by their *total* number of nodes. On top of each bar one can read the fraction value (in black) and the exact numbers (`number of nodes in mGISS` / `number of proper ancestors of Y`) in red. Notice that models with larger numbers of nodes tend to benefit more from our method.

