# OpenReview forum: "The Minimal Search Space for Conditional Causal Bandits"
_ICML.cc/2025/Conference — Submitted to ICML 2025_

### Official Review · Reviewer_1irT · 2025-03-06

**Overall Recommendation:** 3

**Summary:**

his paper explores the problem of minimizing the search space in CB with single-node interventions, covering both do-interventions and conditional interventions. It presents an algorithm for efficiently identifying the minimal globally interventionally superior set, with experiments demonstrating the empirical benefits of intervening solely on the identified sets.

**Claims And Evidence:**

Most likely.

**Essential References Not Discussed:**

1. The paper mentioned soft intervention multiple times, and compares them with conditional interventions. However, prior work on causal bandits with soft interventions (from R1 to R2) is missing, which could provide valuable context and comparisons.

[R1] Varici et al. Causal bandits for linear structural equation models. JMLR.

[R2] Yan et al. Linear Causal Bandits: Unknown Graph and Soft Interventions. NeurIPS 2024

**Experimental Designs Or Analyses:**

The experimental design is unclear to me (see Questions).

**Methods And Evaluation Criteria:**

Yes.

**Other Comments Or Suggestions:**

I think a clearer statement of conditional interventions and the experimental setup would be great, as some aspects remain unclear (see Questions).

**Other Strengths And Weaknesses:**

Strengths

1. Paper looks into minimum search space for single node, which is not done by previous studies.
2. Theoretical results are given through intuitive.

Weaknesses

1. The paper looks into single node intervention without confounders, which seems to be limited and simple. Could the ideas of this paper applied to 2 node intervention and beyond?

2. The two main contributions: (i) the equivalence between conditional and atomic interventions and (ii) the identification of the LSCA closure—seem somewhat separate. I did not seeing the importance of conditional intervention in this paper and it is defined unclear.

**Questions For Authors:**

have a few questions regarding the setting and definition of Conditional Interventions and related concepts:
1. For the policy $g$, is it a fixed function, or does it belong to a set of functions (as Definition 1 states that a policy exists)? Additionally, is the policy (or the set of possible policies) known to the learner when assigning interventions?

2. Is  $\mathbf{Z}_X$ a fixed function, or can it vary when intervening on the same node $X$? Furthermore, is it known to the learner? Does the definition 1 is a superior conditioned on $\mathbf{Z}_X$?

3. What assumptions are made about $g$? For example, for Proposition 4 to hold, it seems necessary that the output range of $g$ matches the do intervention range. Otherwise, if $g(\cdot)=0$, Proposition 4 may become ill-posed.

4. Does the framework assume full intervention capability? That is, can the intervention candidate take any value within the observed range of $X$ when targeting node $X$?

4. The algorithm C4 is unclear to me. Could the authors provide an explanation of how C4 iteratively finds C and $\mathfrak{c}$ using the example in Figure 3?

5. Am I correct in understanding that do-interventions were used in the experiments? Specifically, in the _cond_int_cbn_mab.py file, the do function from the pgmpy package appears to be used. What is the intervention space of MAB, is it binary or discrete or continuous?

**Relation To Broader Scientific Literature:**

The LSCA closure serves as a good supplement to PIMOs in the context of hard interventions in single node intervention.

**Theoretical Claims:**

The conditions/assumptions needed for proposition 4 seems missing while the rest seems correct.

---

> ### Author Rebuttal · Authors · 2025-03-28
>
> Thank you for your review and questions. We are happy to read that you appreciated the intuition given for our theoretical results.
> Your suggestions for definition clarifications will be taken into account in the camera-ready version, if the paper is accepted.
> Further, thank you for pointing out these references which we were not aware of. They seem relevant, and we will carefully consider them and their relationship to our paper.
>
> As a first comment, and to avoid misunderstandings, please note that our work is about single-node conditional interventions, but not about atomic interventions.
> As mentioned in the paper, these results are also valid "for atomic interventions in a **deterministic** causal model". We make no statements about atomic interventions in general causal models.
>
> Furthermore, when it comes to the conditions of Proposition 4, please read our answer to your third question. In summary, Proposition 4 does not need any restrictions on the policies other than what had been stated in the paragraph on conditional interventions in the Preliminaries section.
>
> We will now address the weaknesses and the questions, in the order they appear:
>
> ## Answers to Weaknesses:
> 1. Please note that we do allow for confounders, but not unobserved confounders. We agree that allowing also for unobserved confounders would be an interesting research direction, but it is out of the scope of this already long paper, and would demand significantly more work. See also our answer to question 1 of the reviewer pQQk. Also, as described in the last paragraph of Section 2, the fact that we focus on single-node interventions actually renders our problem significantly complicated (and certainly more complicated than if one allowed for $K > \vert Pa(Y) \vert$ interventions, in which case one would simply need to intervene on all parents of $Y$, as noted by Lee and Bareinboim (2018)). For us, the complexity of our problem comes precisely from allowing single-node interventions and conditional interventions. We also agree that extending this work to K-node interventions with $1 < K < \vert Pa(Y) \vert$ would be an interesting (but again non-straightforward) next research question - please see our answer to question 3 of reviewer pQQk.
> 2. We do not claim an “equivalence between conditional and atomic interventions”, but an “equivalence between the superiority partial orders for conditional interventions and atomic interventions in deterministic causal models”. Furthermore, although one may consider this as an important contribution, the main result is the graphical characterization of the mGISS. This equivalence (Proposition 4) is used to gain intuition about (and also in proving) the graphical characterization of the mGISS, as mentioned in the paragraph after Remark 5. Thus, this equivalence is a step in the direction of proving that the LSCA closure is indeed the mGISS, as explained in the paper.
>    Finally, conditional interventions are important because they better model many real-world situations, as described in the third paragraph of the Introduction and in the references cited therein.
>
> ## Answers to Questions:
> 1. We do not impose any restriction on the policy $g$. It can be any function with the appropriate domain/codomain as described in the Preliminaries section. Accordingly, the learner can choose any policy.
> 2. Yes, $\mathbf{Z}_X$ is fixed for each $X$.
> Also, Definition 1 is indeed stated with respect to a $\mathbf{Z}_X$. Notice however that from Proposition 4 the superiority partial order must be independent from the specific choice of observable conditioning set.
> We can emphasize these points in the paper.
> 3. As mentioned in point 1., no assumptions are needed to make about $g$ other than its domain and codomain being $R_{\mathbf{Z}_X}$ and $R_X$, as written in the Preliminaries. Proposition 4 does not need any  extra restriction on the policies to hold/be well-posed.
> 4. Yes, $X$ can be set to any value in its range. We can emphasize this in the Preliminaries section.
> 5. Covering the entire Figure 3 would be too long for this character-limited rebuttal, so we will focus on the nodes J, K, G, H and E of the graph in Figure 3. Due to line 4 of Algorithm 1, the connectors of J and K are $\mathfrak{c}[J] = J$, $\mathfrak{c}[K] = K$, respectively, since $\mathbf{U} = \\{ J, K \\}$. Now, for node G, the set C (which contains the connectors of the children of G) is $C=\{J\}$, so that $\vert C \vert = 1$ and by lines 8, 9 of the algorithm one has $\mathfrak{c}[G] = J$. Similarly, $\mathfrak{c}[H] = K$. Finally, for node E one has $C = \\{J, K\\}$, so that by lines 10 and 11 of the algorithm $\mathfrak{c}[E] = E$, and $E$ is added to the closure.
> 6. We use conditional do-interventions. Notice that, with pgmpy methods, the context of a conditional intervention is encoded in the "evidence" argument. Also, the MAB intervention space is discrete (and not necessarily binary).

---

> > ### Comment · Reviewer_1irT · 2025-04-04
> >
> > Thanks for the detailed response. However, I’m still a bit confused by the statement *"We do not impose any restriction on the policy $g$"*.
> >
> > For example, the function $g$ could be defined independently of the inputs $\mathbf{Z}_{X}$ so that $do(X= g(\mathbf{Z}_X))=c$, where $c$ can be *any value* within the range. Here I have  some questions.
> > - Is the intervention means two aspects? The target of intervention and the selection of $g$ function
> > - How is the mean reward $\mu_a$ defined, is it a function of $g$ (as it seems need to be a function of $c$)?
> > - How is $g$ selected in C4 algorithm?

---

> > > ### Author Response · Authors · 2025-04-06
> > >
> > > Thank you for your comment.
> > >
> > > Indeed, we can choose $g$ to be a constant function equal to $c$, where $c$ can be any value in the range of $X$.
> > > This is the same as performing the atomic intervention $do(X = c)$. (Are we right to understand that there was a typo in your comment when you wrote $do(X = g(\mathbf{Z}_X)) = c$? We believe you wanted to write $do(X = g(\mathbf{Z}_X) = c)$, which indeed is possible and is just the atomic intervention $do(X = c)$).
> > >
> > > We now address each of your questions:
> > >
> > > 1. Yes, to specify a conditional intervention one needs to select which node $X$ to intervene on, and to choose a policy $g$.
> > > 2. The mean reward $\mu_a$, being the expected value of $Y$ if arm $a$ is selected, depends on the conditional intervention (arm) $a$ that is chosen. Since $a$ is characterized by a node $X$ and a policy $g$ (as described in point 1. above), indeed $\mu_a$ depends on $g$.
> > >    Explicitly, we can write: $\mu_a = \mathbb{E}[Y \mid do(X = g(\mathbf{Z}_X))] = \sum_y p(y\mid do(X = g(\mathbf{Z}_X)) \cdot y$.
> > >    Notice that in the case of an atomic intervention $do(X = g(\mathbf{Z}_X) = c)$ this reduces to the atomic intervention mean reward $\mathbb{E}[Y \mid do(X = c)] = \sum_y p(y\mid do(X = c) \cdot y$.
> > >    We can clarify this in the preliminaries section.
> > > 3. Please note that the C4 algorithm is an algorithm that finds the minimal set of nodes guaranteed to contain the optimal node on which to perform a conditional intervention (that is, the mGISS of the causal graph). It is not the job of the C4 algorithm to select $g$. Instead, $g$ can be learned using a bandits algorithm for conditional interventions. This bandit algorithm can be restricted to only consider conditional interventions on the nodes found by the C4 algorithm. In Figure 4, you can see the impact of restricting a bandits algorithm search to the nodes found by the C4 algorithm on the cumulative regret curves.

---

### Official Review · Reviewer_yhe9 · 2025-03-21

**Overall Recommendation:** 4

**Summary:**

# Summary
- This paper is refreshingly nice paper to read, and the authors have taken care to make the paper easily readable. For instance, I am comparing with general papers I read in this area.
  - In that sense, it is similar to papers by Lattimore which rate high for clarity.

  - For example, instead of simply stating theorems, the authors provide intuitions behind their claims.

- The authors take the problem of causal bandits where interventions are conditional. Unlike prior work where the conditioning gives rise to separate graphs, this is closer to Bareinboim's 2019 work where they define the SCM-MAB problem, wherein the interventions are single node conditional interventions.

**Claims And Evidence:**

(a) we establish a graphical characterization of the minimal set of nodes guaranteed to contain the optimal node on which to perform a conditional intervention;
- Yes this is shown in Section 4.

(b) we propose an algorithm which finds this set, given only the causal graph, with a time complexity of O(|V | + |E|).
- Yes this is evidenced and correct.

**Essential References Not Discussed:**

The authors may want to add references to recent works in minimal intervention sets over MECs. This is not exactly the same problem as the authors are trying to address, but is related.

**Experimental Designs Or Analyses:**

The authors have performed experiments on graphs derived from real world datasets. So there is no intrinsic bias towards their algorithm through instance selection.

**Methods And Evaluation Criteria:**

I especially liked the fact that they chose to gather their graphs from real-world datasets. The number of nodes may go up to 20, which seems reasonable given the claim of extending to real world applications.

**Other Comments Or Suggestions:**

- The notations with many superscripts and subscripts is at times confusing. In the final manuscript, either simplifying the notations, or adding a table of notations would be useful in case the authors have sufficient space for the same. But i leave this to the authors best judgement.

**Other Strengths And Weaknesses:**

# Strengths
1. High clarity of writing.
2. This generalizes previous works in the area.

# Weaknesses:
1. The graph does not contain any latents. As the authors point out, extending to unobserved confounders would be an interesting direction of future work.

**Questions For Authors:**

- Does the formulation easily extend to the simple regret setting?
- What if contexts were vector spaces vs discrete spaces? how do these tie in? Is this an interesting direction?

**Relation To Broader Scientific Literature:**

The authors position their work well with respect to recent literature in the area of causal bandits. Many relevant related works have been discussed.

**Theoretical Claims:**

Their theoretical claims were proved to the best of my reading. I went through the main paper, as well as parts of the appendix.

---

> ### Author Rebuttal · Authors · 2025-03-28
>
> Thank you for your review and questions. We are glad that you enjoyed reading our paper and that you appreciate our efforts to make the message clear.
>
> The only paper we were able that seemingly matches your suggestion to add "references to recent works in minimal intervention sets over MECs" was a technical report from February 2025 by Park, Arditi, Bareinboim and Lee, "Structural Causal Bandits under Markov Equivalence". Is this what you meant or did you have other papers in mind that we missed?
>
> About the weakness you mentioned, although we agree that the cases with latent confounding and K-node interventions would be an interesting research direction, each of these topics would, from our perspective, require substantial extra research exceeding the scope of this already long paper, and may be the focus of future research.
>
> We will do our best to simplify the notation where possible, and will consider adding a notation table to the appendix.
>
> We will now address the questions, in the order they were posed.
>
> ## Answers to Questions:
> 1. Yes, we expect it to also hold for simple regret. Note that the set mGISS that we found is guaranteed to contain the node with the best conditional intervention (best arm), and we expect that, if it is easier for the algorithm to find the best arm, then it is more likely that the regret of the chosen arm at the end of training is indeed the smallest possible.
>
> 2. We focused on the case where variables have discrete (and finitely many) values since we are interested in the bandits problem (with finitely many arms). If there were an uncountable amount of possible interventions on a node, one would need to use other frameworks, such as Bayesian Optimization. We do expect our work to be amenable to adaptation to that scenario. So yes, continuous/vector space variables (both to intervene on and as contexts) are an interesting research direction.

---

### Official Review · Reviewer_pQQK · 2025-03-24

**Overall Recommendation:** 4

**Summary:**

This paper studies the conditional causal bandit problem, where interventions depend on observed variables rather than being fixed. It provides a graphical characterization of the minimal set of nodes that guarantees the presence of the optimal conditional intervention. An efficient algorithm with O(|V| + |E|) complexity is proposed to identify this minimal set using only the causal graph. The correctness of both the characterization and algorithm is formally proven. A key result shows that the same minimal set applies to deterministic causal models with atomic interventions. Empirical results demonstrate that the method significantly prunes the search space in both synthetic and real-world graphs, especially in large, sparse models. Integrating this approach into multi-armed bandit (MAB) algorithms improves efficiency and speeds up convergence. These contributions provide a foundation for more efficient decision-making in causal bandits with conditional interventions.

**Claims And Evidence:**

Yes, all claims in the submission are supported by clear and convincing evidence. All the theorems and propositions have been formally stated and proved in the supplementary material.

**Essential References Not Discussed:**

The paper cites and discusses all necessary and relevant prior work.

**Experimental Designs Or Analyses:**

The experiments appear to be sound and demonstrate that the proposed algorithm significantly prunes the search space and substantially accelerates convergence rates when integrated into standard multi-armed bandit algorithms.

**Methods And Evaluation Criteria:**

The authors conduct experiments to assess the fraction of the search space that can be expected to be pruned using the proposed method in both randomly generated and real-world graphs. Additionally, they demonstrate, using well-known real-world models, that their intervention selection can significantly improve a classical MAB algorithm.

**Other Comments Or Suggestions:**

I did not see any major typos or other related issues in the manuscript.

**Other Strengths And Weaknesses:**

I have listed the main strengths and contributions of the paper in the Summary section and the section "Relation to Broader Scientific Literature." In terms of main weaknesses, some assumptions restrict the practical applicability of the work, including no latent confounder assumption, and considering single-node interventions only. The work also assumes that the conditioning set is already known to the agent, whereas in many cases, it may not be clear what conditions to use in advance.

**Questions For Authors:**

I have some questions for the authors:

* The authors assume that there are no latent variables and leave the extension of this work to the case where latent variables are present as future work. The work by Lee and Bareinboim (2018), which characterizes the set of possibly optial arms when confounders are present. Is it possible to use insights from this work to relax this assumption in the current setup. Could the authors elaborate on this?

* Also, the causal graph needs to be known for the proposed approach. Suppose the causal graph is unknown—can one use targeted causal discovery methods, like in the paper https://arxiv.org/pdf/2301.11401, to relax this assumption in the current setup?

* The authors focus on single-node interventions. What about multi-node interventions, say up to K nodes, simultaneously? Is it possible to use a similar approach in this case?

* In the experimental section, the authors use a per-context UCB-style algorithm. Is it possible to transfer information across contexts and accelerate the learning process?

**Relation To Broader Scientific Literature:**

This paper introduces a novel approach to causal bandits by focusing on conditional interventions, which allow the value of the intervened variable to depend on the observed values of other variables. Unlike previous work on hard interventions, where variables are set to fixed values, this method provides a more flexible and realistic framework for decision-making in various applications such as healthcare and personalized recommendations. The authors present a graphical characterization of the minimal set of nodes needed to find the optimal conditional intervention, along with an efficient algorithm for identifying this set. Their contributions include reducing the search space for optimal interventions and demonstrating substantial improvements in convergence rates when integrated into standard bandit algorithms. This work addresses a gap in the literature by fully characterizing the minimal search space for single-node conditional interventions, offering a new perspective on intervention selection in causal models.

**Theoretical Claims:**

I have skimmed through the proofs in the supplementary material for the main theoretical results, and it appears to be sound.

---

> ### Author Rebuttal · Authors · 2025-03-28
>
> Thank you for your review and questions. We are glad to read that you found our arguments clear and convincing.
>
> About the weaknesses that you mentioned, we agree that the cases with latent confounding and K-node interventions would be an interesting research direction.
> From our perspective, each of these topics would require substantial extra research exceeding the scope of this already long paper, and may be the focus of our future research.
> Please note that, although the conditioning set needs to be known when performing a conditional intervention, it does not need to be known to establish the minimal search space - from Proposition 4, the superiority partial order must be independent from the specific choice of observable conditioning set.
>
> We will now address the questions.
>
> ## Answers to Questions:
> 1. Yes, we believe it may be possible to adapt parts of the work by Lee and Bareinboim (2018) the the single-node, conditional intervention case, but we do not see a straightforward way to do it. In particular, we expect a notion related to their concept of interventional border to be useful in our case, but exactly how would need substantial further research.
>
> 2. Yes, we do not see a problem with combining such causal discovery methods in order to tackle problems where the causal graph is unknown in order to allow the use of our method. We can make this explicit in our Discussion/Conclusion section.
>
> 3. Yes, we expect that a similar approach would be possible for the K-node intervention case, but not in a trivial way: for example, we do not expect the minimal search space to simply be the cartesian product of the LSCA closures, as one could at first suspect.
>
> 4. Possibly, but that would require us to make assumptions about the connection between said contexts. We took the general approach of not making any such assumptions.

---

> > ### Comment · Reviewer_pQQK · 2025-04-01
> >
> > I have read the rebuttal and agree that the points I mentioned in weaknesses and questions can be addressed in future work. Based on the contributions in the paper and rebuttal, I will maintain my acceptance score.

---

### Decision · Program_Chairs · 2025-05-01

**Decision:**

Reject

**Comment:**

The paper studies conditional causal bandits, a multi-armed bandit (MAB) with a causal structure among the variables, where the actions are conditional interventions over single nodes. It proposes using the causal graph to reduce the action space and then running a standard MAB algorithm. To that end, it presents a graphical characterization of the minimal set of nodes containing the optimal action, thereby reducing the search space for optimal actions. It also presents an algorithm to find this minimal set given the causal graph with a time complexity linear in the number of nodes and edges.

The paper’s contribution toward reducing the action space in the setting with conditional interventions over single nodes is novel and valuable. The empirical results support that this search space reduction lowers the cumulative regret, as expected. Nevertheless, the proposed method treats the actions independently in the bandit algorithm, not utilizing the causal information in that phase. The reviewers have pointed out some limitations and raised some questions, part of which are addressed. They have also recommended interesting extensions to the work which the authors could consider in future submissions.

The paper has major drawbacks, especially in its presentation, formalism, and empirical studies, that undermine the motivation and significance of its contributions.

1) The studied setting is not well-motivated. In the introduction, the paper motivates the conditional intervention for recommendation systems and medical treatments, mostly by referring to papers on contextual bandits. In conventional contextual bandits, the context variables are fixed while here, the context variables change according to the choice of the intervention node. Therefore, motivating scenarios/applications tailored to this setup is essential.
2) The connection to contextual bandits is missing. Given the long-standing area of contextual bandits, the paper needs to clearly discuss the fundamental similarities and differences between the studied setup and that of contextual bandits (not just the contextual causal bandit works mentioned in the “Related Work”).
3) The complexity of the decision-making due to conditional interventions is obscure throughout the paper since their description is unclear, e.g., the role of Z_X and g, as Reviewer 1irT pointed out. The rebuttal clarifies some of the ambiguities but poses concerns that this complexity is not sufficiently addressed or discussed. For instance, the number of decisions that the MAB must consider per intervention node X scales exponentially with respect to the number of variables in Z_X. The paper does not restrict Z_X other than the necessary requirements and treats the resulting decisions independently. Therefore, it is crucial to explicitly state this complexity. Additionally, following Reviewer pQQK’s suggestion, the authors may be able to leverage the relation among the contexts to even reduce this complexity.
4) The empirical setups do not match the motivation of the proposed setup with conditional interventions, and they lack details, as Reviewer 1irT pointed out. I recommend that the authors create/add a relatively realistic example, present that early in the paper to motivate the necessity of conditional interventions, and run their experiments on that example as well.
5) The presentation of the paper, particularly the “Introduction” (the third paragraph) and the “Related Work” (the third paragraph), seems to suggest that it is better to use the causal model to reduce the search space rather than leveraging the causal relationships in a bandit algorithm to accelerate learning. This would be a misleading argument as these are complementary approaches to tackle causal bandits. To optimally utilize the causal model for decision-making, one would want to reduce the search space a priori if possible and then leverage the causal relationships throughout the bandit algorithm. Even if the paper’s contribution relates to only one of these aspects, the narrative should reflect the utility of both. Besides, I recommend that the authors consider using those causal bandit methods (some of which are cited in the paper) instead of independent UCB, although I understand that requires some modifications to work for this setting with conditional interventions.